# The effect of rainfall variability on Nitrogen dynamics

# in a small agricultural catchment

- Qiaoyu Wang<sup>1,3</sup>, Jie Yang<sup>1,2</sup>, Ingo Heidbüchel<sup>4,5</sup>, Teng Xu<sup>1,3</sup>, Chunhui Lu<sup>1,2</sup>
- <sup>1</sup>The National Key Laboratory of Water Disaster Prevention, Hohai University,
- Nanjing, China<sup>2</sup>College of Hydrology and Water Resources, Hohai University,
- Nanjing, China

- <sup>3</sup>College of Water Conservancy and Hydropower Engineer, Hohai University,
- Nanjing, China
- <sup>4</sup>UFZ Helmholtz-Centre for Environmental Research GmbH, Department of
- Hydrogeology, Leipzig, Germany
- <sup>5</sup>Hydrologic Modeling Unit, Bayreuth Center of Ecology and Environmental
- Research (BayCEER), University of Bayreuth, Bayreuth, Germany
- Correspondence to: Jie Yang (yangj@hhu.edu.cn); Chunhui Lu (clu@hhu.edu.cn)

Abstract. Throughout history, extreme storms and droughts have had serious impacts on society and ecosystems globally. Rainfall variability particularly has been identified as a primary manifestation of climate change. However, so far little has been done to explore the effect of rainfall variability on water quality. This study aims to investigating the effect of rainfall variability on nitrogen (N) dynamics and its potential influence on water quality. The transport of water and nitrate was simulated for a small agricultural catchment in Central Germany using the fully coupled surface-subsurface model HydroGeoSphere. Rainfall time series with specific climatic characteristics were generated using a stochastic rainfall generator. N transformation and transport were compared for four scenarios (with high, normal, low annual precipitation, and low annual precipitation coupled with reduced plant uptake, respectively) in order to identify the impact of inter-annual rainfall variability. The results suggest that higher annual precipitation enhances N transformation and transport, whereas lower annual precipitation is conducive to N retention capacity. Nonetheless, the retention capacity declines severely when vegetation suffers from drought stress, suggesting that vegetation plays a vital role in N dynamics under extreme droughts. The linear regressions between selected parameters of the rainfall generator and N loads/fluxes were analyzed to elucidate the impact of intra-annual rainfall variability. The results indicate that wet/dry conditions and different dry-wet patterns caused by storm duration distributions and inter-storm period distributions can significantly affect N loads and in-stream nitrate concentration, respectively. In the warm season, droughts prompt the accumulation of soil organic nitrogen (SON), but drying-wetting cycles can enhance the extensive transformation of SON. In-stream nitrate concentration dramatically elevates during the rewetting period after the drought. Wet/dry conditions and patterns of precipitation intensity within a year determined by seasonal average rainfall intensity and the probability of drizzle events prominently alter mineralization and plant uptake. Elevated mean rainfall intensity has merely a small effect on stream water quality. Therefore, mineralization and plant uptake are critical processes governing N dynamics and the impact of nitrate on water quality under varying rainfall conditions. Overall, the study clarifies the effect of rainfall variability on N dynamics in a small agricultural catchment, which provides theoretical support for formulating fertilization strategies and protecting aquatic ecosystems under climate change.

14

1516

17

18

19

2021

3435

39

44

**Key Points:** N dynamics, HydroGeoSphere, Rainfall variability, Stochastic rainfall generator

## 49 1 Introduction

The hydrological processes are susceptible to meteorological conditions on various spatial and temporal scales [Ionita et al., 2017; Laaha et al., 2017; Zhang et al., 2021]. In the past decades, extreme climate events intensified by human-induced climate change have frequently occurred globally [Pall et al., 2011; Min et al., 2011; Williams et al., 2015; Hari et al., 2020], most of which caused water scarcity and poor water quality at regional scales [Zwolsman and van Bokhoven, 2007; Delpla et al., 2009; Whitehead et al., 2009; Stahl et al., 2016; Ballard et al., 2019; Bauwe et al., 2020; Geris et al., 2022]. Heavy rainstorms and severe droughts being the predominant extreme climate events around the globe share the common characteristic of rainfall variability [Trenberth et al., 2011; Pendergrass et al., 2017; Hanel et al., 2018]. In the context of global warming scenarios, anthropogenic amplification of rainfall variability has been identified [Zhang et al., 2024]. Thus, the effect of rainfall variability on water resources has attracted much attention around the world. During heavy rainstorms, extraordinary rainfall amounts and intensities cause large amounts of rainwater to infiltrate into soils and trigger flash floods in a short time. Increased groundwater flow and enhanced surface runoff stimulate the movement of solutes retained in the soil, which thereby can lead to water quality degradation [Geris et al., 2022]. Different from heavy rainstorms, severe droughts driven by precipitation deficits need to take several months or potential years to reach their full intensity

[Otkin et al. 2018], from which it takes 1-2 years for hydrological components to recover [Hanel et al., 2018]. Wilusz et al. [2017] decomposed the relationship between rainfall variability and transit times, a reflection of water velocities that control solute transport, and illustrated that climate change may seasonally alter the ages of water in streams and thereby influence water quality in the future.

Nitrate (NO<sub>3</sub>-N) is a major solute threatening the quality of drinking water and destroying the structure and functions of aquatic ecosystems [Vitousek et al., 2009; Alvarez-Cobelas et al., 2008; Dupas et al., 2017]. The nitrate turnover processes established at the catchment scale are expected to change due to climate change [Whitehead et al., 2009; Hesse and Krysanova, 2016; Mosley, 2015], especially due to increasing drought events [Zwolsman and van Bokhoven, 2007; Ballard et al., 2019; Zhou et al., 2022; Winter et al., 2023]. Several extreme rainfall and droughts occurred in Central Europe during the last two decades, which has attracted widespread attention [Ulbrich et al., 2003; Fink et al., 2004; Orth, Vogel, Luterbacher, Pfister, and Seneviratne, 2016; Thieken et al., 2016; Hanel et al., 2018; Hari et al., 2020; Voit, P. and Heistermann, M., 2024]. Notably, the 2018 event triggered unprecedented tree mortality across multiple species in Central European forests, accompanied by unexpectedly persistent drought legacy effects [Schuldt et al., 2020]. Zhou et al. [2022] detected higher soil N surplus (total N input with the crop/plant uptake subtracted) and decreased the terrestrial N export in agricultural areas located in Central Germany

during the drought years (2015-2018). The same phenomenon reported in the Nitrate Report 2020 of the Netherlands (RIVM, 2021) indicates that more N was retained in the soil during the drought period compared to the pre-drought period. However, by studying the 2018-2019 (consecutive) drought in Central Germany, Winter et al. [2023] drew the conclusion that severe multi-year droughts can reduce the nitrogen (N) retention capacity of catchments. They seem opposite conclusions, which can be attributed to the different investigation timescales. The latter study considered the subsequent rewetting period, when most nitrogen accumulated during the drought left the catchment. Leitner et al. [2020] also found that in the year after a summer drought, NO<sub>3</sub>- leaching via soil water seepage was significantly elevated compared to the long-term mean in a temperate mixed forest on karst, which was investigated in wetland-influenced catchments as well [Watmough et al., 2004]. These prove that rainfall variability has a profound impact on N dynamics. Therefore, it is imperative to shed light on the impact of rainfall variability on water quality in terms of N dynamics.

To fill the gap, the present study explored the impact of rainfall variability on N dynamics and its potential influence on water quality across inter-annual and intra-annual timescales. To characterize rainfall variability, a stochastic rainfall generator [Robinson and Sivapalan, 1997] was employed to generate rainfall time series with different climatic characteristics. The research was conducted in a small

agricultural catchment located in Central Germany, where a hydrological model was previously established utilizing the fully coupled surface-subsurface numerical simulator HydroGeoSphere [Yang et al., 2018]. The framework of N dynamics was modified from the ELEMeNT approach [Exploration of Long-tErM Nutrient Trajectories; Van Meter et al., 2017]. The research is divided into two main components. First, three representative years (with high, normal, and low annual precipitation amounts, respectively) were chosen from the past two decades in Central Germany as the target scenarios. A fourth scenario with low annual precipitation amounts coupled with reduced plant uptake represents a case where vegetation is partially destroyed by extreme droughts. The statistical analyses of N loads and fluxes and a comparison across different scenarios were conducted to reveal the influence of inter-annual rainfall variability. Second, rainfall time series generated using the stochastic rainfall generator by separately altering specific parameters were used to substitute the rainfall data in the simulation period to drive the flow and nitrogen transport models. The responses of the N loads and fluxes to the parameters (e.g., the amplitudes of the seasonal variations in the storm duration and inter-storm period) were thoroughly analyzed to clarify the effect of intra-annual rainfall variability. The study will provide theoretical support for formulating fertilization strategies and protecting aquatic ecosystems in the context of climate change.

## 2 Data collection

### 2.1 Study Site

The studied catchment, Schäfertal, is located on the lower reaches of the eastern Harz Mountains, Central Germany, with an area of 1.44 km<sup>2</sup>. Since 1968, this first-order catchment has been subject to broad hydrogeological investigations, analyses, and research [Altermann et al., 1970; Altermann et al., 1977; Borchardt et al., 1981; Altermann et al., 1994; Wollschläger et al., 2016]. The valley bottom contains riparian zones with pasture and a small channel (Figure 1a). The hillslopes on both sides of the channel have an average slope of 11°, mostly used as farmland. The farmland undergoes intensive agriculture, primarily winter wheat growing [Yang et al., 2018]. Two small portions near the western edge are mostly forest. The types of land use in the catchment do not generally convert until the economic and ecological goals vary between years, e.g., shifting between planting and pasture [Wang et al., 2023]. A meteorological station, which is 200 m from the catchment outlet, provides records of precipitation, air and soil temperatures, radiation, and wind speed. The catchment outlet in the stream, where a gauge station was built, is considered as the unique exit that allows water and solute to leave the catchment and enter the downstream catchments. This is because a subsurface wall (~55 m long and ~7 m deep) was erected across the valley to block subsurface flow. The gauge station provides discharge data at 10-minute intervals, aggregated to daily means in this study. Nitrate

concentration data were measured by sampling near the gauge station at 14-days to monthly intervals [Dupas et al., 2017], covering the period 2001–2010.

The aquifer is thin, with the thickness varying from  $\sim$ 5 m near the valley bottom to  $\sim$ 2 m at the top of the hillslopes (I-I cross section, Figure 1a). Thirteen wells, each ~2 meters deep, were constructed and fitted with automated data loggers to record groundwater levels. The groundwater levels exhibit pronounced seasonal variations, rising to the land surface during winter and receding to depths of ~3 meters below the ground during summer. The groundwater storage is low (~500 mm, stored water volume divided by catchment area). Most of the groundwater converges towards the channel vicinity, with the upper sections of the hillslopes typically in an unsaturated state [Yang et al., 2018]. Luvisols and Gleyic Cambisols are the aquifer materials of the hillslope. The valley bottom is dominated by Gleysols and Fluvisols (Figure 1b) [Anis and Rode, 2015]. Generally, the aguifer is comprised of two horizontal layers: a top layer of approximately 0.5 m thickness, with higher permeability, higher porosity, and the developed root zone from crops; and a base layer with less permeability due to the high loam content [Yang et al., 2018]. The bedrock underlying the aquifer is comprised of greywacke and shale [Graeff et al., 2009]. Owing to the aquifer material and unknown hydraulic properties, the bedrock is often regarded as impermeable.

**Figure 1.** (a) The catchment 'Schäfertal' indicated by the red line (background image from © Google Maps), with a cross-sectional view for the flow and saturation [*Yang et al.*, 2018]. (b) The distribution of soil type in the catchment. (c) The measured daily precipitation (P), discharge (Q), and the simulated actual evapotranspiration (ET) [*Yang et al.*, 2018].

## 2.2 Measured data

The studied catchment exhibits a temperate and humid climate with pronounced seasonality. The humid climate in wet regions is quantified by an aridity index of 1.0. The ET is the main driver of the hydrologic seasonality, as the precipitation is uniformly distributed across the year. According to the meteorological data records

from 1997 to 2010, the mean annual precipitation (P) and discharge (Q) are 610 mm and 160 mm, respectively (Figure 1c). Based on the 14-year water balance (P = ET + Q), the actual mean annual evapotranspiration (ET) is 450 mm. The in-stream nitrate concentration ( $C_Q$ ) was measured at fortnightly to monthly intervals [Dupas et al., 2017], covering the period 2001-2010. The N surplus, which is the annual amount of nitrogen remaining in the soil after the consumption by plant uptake from the external N input, was estimated as 48.8 kg N ha<sup>-1</sup> year<sup>-1</sup> during 1997–2010 for this catchment [Bach and Frede, 1998; Bach et al., 2011]. The  $C_Q$  and N surplus are used to calibrate the N transport model [Wang et al., 2023]. Adequate data from numerous investigations and previous research supports the exploration of N dynamics in the agricultural catchment.

## 3 Methodology

The hydrological model of Schäfertal catchment was established using HydroGeoSphere [Therrien et al., 2010] in the previous study [Yang et al., 2018]. The framework of N dynamics [Yang et al., 2021] modified from the ELEMeNT modeling approach [Exploration of Long-tErM Nutrient Trajectories, Van Meter et al., 2017] was applied to track the fate of N in soil and groundwater in the present study. Based on the hydrological and transport model, the effect of topographic slope on the export of nitrate [Yang et al., 2022] and the spatial-temporal variation of nitrogen retention [Wang et al., 2023] were explored. In the present study, the two models were

employed to investigate the effect of rainfall variability on N dynamics, for which rainfall time series with climatic characteristics, substituting for the precipitation data during the simulation period, were generated by a stochastic rainfall generator [Robinson and Sivapalan, 1997; Wilusz et al., 2017]. These models are described below.

#### 3.1 Flow modeling

HydroGeoSphere is a 3D control volume finite element simulator that can model fully coupled surface-subsurface hydrological processes by the dual nodes approach. It can not only describe crucial hydrological processes such as dynamical evapotranspiration, snow, sublimation, snowmelt, and freeze and melt of pore water, but also simulate 2D overland flow by Manning's equation and the diffusive-wave approximation of the St. Venant equations, 3D variably saturated underground flow by Richards' equation and Darcy's law, flow in porous media, as well as the transport of reactive solutes. In addition, HydroGeoSphere allows the simulation of 1D surface flow in a channel network and water exchange flux between the channel domain and the subsurface domain [Yang et al., 2015]. HydroGeoSphere has been frequently used to model catchment hydrological processes and solute transport in many previous studies [e.g., Therrien et al., 2010; Yang et al., 2018]. Please refer to Therrien et al. [2010] for the governing equations and technical details.

The hydrological model of Schäfertal catchment is briefly recapitulated in the

following. More details are provided in Yang et al. [2018]. In the model, the subsurface domain between the surface and bedrock was depicted using 3D prisms with side length ranging from 30 m to 50 m. The surface domain was filled by the uppermost 2D triangles of the generated 3D prismatic mesh, while the channel was delineated by 1D segments with specified widths and depths. According to different aguifer materials and the difference in their permeability in the vertical direction, the subsurface domain was separated into ten property zones with zonal hydraulic conductivity and porosity values. The surface domain and channel were parameterized with a Manning roughness coefficient representing the land use. Spatially uniform and temporally variable precipitation (Figure 1c) was applied to the entire surface domain. ET is computed as a combination of plant transpiration from the root zone and evaporation down to the evaporation depth [Therrien et al., 2010]. A critical depth boundary condition was assigned to the outlet in the channel domain to simulate the discharge (Q) of the catchment. In order to eliminate the influence of initial conditions, a preliminary model run was performed for the period spanning 1997 to 2007. The simulated results at the end of the period were taken as initial conditions for the actual simulations. Key parameters that significantly influence the hydrological processes were selected for the calibration [Anis and Rode, 2015; Graeff et al., 2009] using the software package PEST [Doherty and Hunt, 2010]. PEST uses the Marquardt method to minimize a target function by varying the values of a given set of parameters until the

optimization criterion is reached. The groundwater levels measured in groundwater wells and the discharge measured at the gauging station were used as target variables. Considering high CPU time (~1 day to run the model for the period from 1997 to 2007), the calibration period spans 1 year, from October 2002 to October 2003. Subsequently, the calibrated model was verified by reproducing time-variable groundwater levels for the wells over the entire simulation period [Yang et al., 2018].

#### 3.2 Nitrogen transport in soil and groundwater

The transformation and transport of nitrogen in the underground area are tracked by the framework of N dynamics modified from ELEMeNT. ELEMENT is a comprehensive model maintaining a landscape memory, that is, considering the effect of not only current conditions but also past land use and nutrient dynamics on current fluxes [Van Meter et al., 2017]. External nitrogen input goes through transformation and transport in the soil, and subsequently filters into groundwater, and gets exported to the surface water body. Throughout the nitrogen cycle, various forms of nitrogen undergo complex biogeochemical processes.

The framework includes a N source zone forming in shallow soil and a groundwater zone (Figure 2). There are two assumptions in the N source zone: 1) the total N load in the N source zone is comprised of an organic N (SON) pool and an inorganic N (SIN) pool; 2) the external N input contributes only to the SON. The external N input represents atmospheric deposition, biological fixation, animal manure from the

pasture area, and N fertilizer from the farmland. The SON is further distinguished as active SON with faster reaction kinetics and protected SON with slower reaction kinetics. Both SON(a) and SON(p) are transformed into SIN by mineralization. SIN is further consumed by plant uptake and denitrification, and leaches from soil to groundwater as dissolved inorganic N (DIN), representing mainly nitrate in the studied catchment [Yang et al., 2018; Nguyen et al., 2021]. DIN can further undergo denitrification until being exported to the stream. The framework is able to capture the main processes of nitrogen transformation and transport in soil and groundwater [Yang et al., 2018].

**Figure 2.** The framework simulating the transformation and transport of nitrogen in soil N source zone and groundwater zone, modified based on [*Yang et al., 2021*].

The governing equations to calculate N fluxes follow the ones of the framework in Yang et al. [2021]. The land-use dependent protection coefficient h [Van Meter et al.,

- 2017] determines the amount of external N input that contributes to the protected
- SON (SON(p)), and the residual contributes to the active SON (SON(a)).
- Mineralization and denitrification are described as first-order processes. Based on the
- results of Wang et al. [2023], the effect of wetness is considered in mineralization,
- using:

$$MINE_a = k_a \cdot f(temp) \cdot f(wetness) \cdot SON_a$$
 (1)

$$MINE_p = k_p \cdot f(temp) \cdot f(wetness) \cdot SON_p$$
 (2)

$$278 DENI_S = \lambda_S \cdot SIN (3)$$

$$279 DENI_g = \lambda_g \cdot DIN (4)$$

- where  $MINE_a$ ,  $MINE_p$  (kg ha<sup>-1</sup> day<sup>-1</sup>) are the mineralization rates for active SON
- and protected SON, respectively.  $DENI_s$  and  $DENI_g$  (kg ha<sup>-1</sup> day<sup>-1</sup>) are the
- denitrification rates for SIN and DIN, respectively.  $k_a$ ,  $k_p$ , and  $\lambda$  (day<sup>-1</sup>) are
- coefficients for the three first-order processes. f(temp) and f(wetness) are
- factors representing the constraints by soil temperature and wetness [Lindström et al.,
- 2010; Wang et al., 2023], respectively. Plant uptake rate UPT follows the equation
- used in the HYPE model [Lindström et al., 2010]:

$$287 \quad UPT = min (UPT_P, 0.8 \cdot SIN) \tag{5}$$

$$UPT_{P} = p1/p3 \cdot (\frac{p_{1}-p_{2}}{p_{2}}) \cdot e^{-(DNO-p_{4})/p_{3}} / (1 + (\frac{p_{1}-p_{2}}{p_{2}}) \cdot e^{-(DNO-p_{4})/p_{3}})^{2}$$
 (6)

where *UPT* and *UPT*<sub>P</sub> (kg ha<sup>-1</sup> day<sup>-1</sup>) are the actual and potential uptake rates,

respectively. The logistic plant growth function is considered in the equation of 291 potential uptake rates. DNO is the day number within a year. p1, p2, p3 are three 292 parameters depending on the type of plants, whose units are (kg ha<sup>-1</sup>), (kg ha<sup>-1</sup>), and 293 (day), respectively. p4 is the day number of the sowing date. The leaching rate adapts 294 a first-order process considering soil saturation and groundwater velocity, using:

$$295 LEA = k_l \cdot f_w \cdot f_q \cdot SIN (7)$$

$$296 f_w = \frac{S - S_r}{1 - S_r} (8)$$

$$297 f_q = MIN(\frac{q}{q_{ref}}, 1) (9)$$

where LEA is the leaching flux of SIN from the N source zone to the groundwater,  $k_l$  is a leaching coefficient (day-1),  $f_w$  and  $f_q$  are two factors representing the 299 300 constraints of soil saturation and groundwater velocity to the leaching process, respectively. S is the soil saturation, and  $S_r$  is the residual saturation. q (m day<sup>-1</sup>) is 301 the groundwater Darcy flow rate,  $q_{ref}$  (m day-1) is a reference Darcy flow rate. This 302 formulation of LEA is modified from the one used in Van Meter et al. [2017]. 303 304 In these hydrogeochemical processes, a portion of N is retained in the catchment as 305 the biogeochemical legacy in soil or the hydrological legacy in groundwater or 306 permanently leaves catchments by denitrification, which does not degrade the water 307 quality of the catchment during a certain period. Therefore, the N retention is used to

bodies during a certain period [Wang et al., 2023], which is the fraction of the N

quantify a catchment's capacity to prevent nitrogen from entering surface water

retained in the catchment and consumed via denitrification to the total external N input, calculated using [Ehrhardt et al., 2021]:

Retention = 
$$1 - \frac{N_{\text{out}}}{N_{\text{in}}} = 1 - \frac{\int_{t_1}^{t_2} N_{\text{outlet}}}{\int_{t_1}^{t_2} N_{\text{input}}}$$
 (3)

- where the  $N_{outlet}$  is the N mass leaving the catchment through the outlet during the time period  $(t_1 t_2)$ .
- Due to the lack of spatiotemporal variation information of the external N input, its 315 value was fixed at 180 kg ha<sup>-1</sup> year<sup>-1</sup> according to Nguyen et al. [2021], where the N 316 balance was simulated in the upper Selke catchment covering the Schäfertal 317 318 catchment. The protection coefficient h was selected as 0.3 according to the values 319 reported in Van Meter et al. [2017]. p4 as the sowing date of plant growth activities 320 was set to 63 days (early March) [Yang et al., 2018]. The DIN was transported in the 321 coupled surface water-groundwater system, with longitudinal and transverse 322 dispersion coefficients of 8 and 0.8 m, respectively. Other parameters relative to N 323 dynamics were calibrated by PEST in empirical ranges. The measured C<sub>0</sub> at the gauge station and the N surplus of 48.8 kg N ha<sup>-1</sup> year<sup>-1</sup> were used as the target variables. 324 The calibration period spans from March 2001 to August 2003, during which 325
  - The N source zone serves as a boundary condition at the top of the aquifer for simulating DIN transport in the groundwater. The bedrock is treated as impermeable

successive C<sub>Q</sub> data was obtained. The CPU time was ~2 h for a single iteration.

for water and nitrate. The catchment outlet is the only boundary allowing the exit of nitrate from the catchment. In order to minimize the influence of the initial conditions, a preliminary transport simulation (together with the flow simulation) was performed with zero load in the SON and SIN pools and zero DIN concentration in the catchment, such that a quasi-steady state for the SON and SIN pools can be reached at the end of the preliminary simulation. The resulting data (N loads and concentrations) were used as initial conditions for the actual simulations.

#### 3.3 Stochastic rainfall generator

In order to investigate the effect of rainfall variability on N dynamics, rainfall time series with different climatic characteristics were generated by a stochastic rainfall generator. The stochastic rainfall generator is a stochastic model of rainfall time series originally created to investigate the timescales of flood frequency response to rainfall, incorporating any combinations of storm, within-storm as well as between-storm, and seasonal variabilities of rainfall intensity [Robinson and Sivapalan, 1997]. It can output rainfall series representing different rainfall patterns under climate change. The equations can be found in Section S1 of the Supporting Information.

We adopted the source code in Python v2.7 of the rainfall generator from Wilusz et al. [2017]. The stochastic model derives monthly average storm durations and inter-storm periods from seasonal parameters: averages  $(\gamma_s, \delta_s)$ , amplitudes  $(\alpha_{\gamma}, \alpha_{\delta})$ , and phases  $(\tau_{\gamma}, \tau_{\delta})$ .  $a_1^1 \sim a_1^4$ ,  $b_1^1 \sim b_1^4$ ,  $a_2$  and  $b_2$  are the parameters characterizing

| the dependence of average rainfall intensity on storm duration for the first season (Jan           |
|----------------------------------------------------------------------------------------------------|
| - Mar), the second season (Apr - June), the third season (Jul - Sep), and the fourth               |
| season (Oct - Dec). In addition, there is an isolated parameter P <sub>drizzle</sub> in the source |
| code, the probability of drizzle events. In the model, storms with precipitation smaller           |
| than 4mm are drizzle events, while storms with precipitation greater than 4mm can be               |
| identified as synoptic frontal events.                                                             |

- Rainfall time series representing different climate conditions were generated following these steps below:
- Three representative years were selected from historical meteorological data, which are the wet year (2007, P = 916.3 mm), the normal year (2008, P = 588.7 mm), and the dry year (2018, P = 444.1 mm);
  - (ii) For each of the representative years, a set of parameters was determined so that the generated rainfall time series can fit the actual rainfall data of this year best. This inverse process was conducted using PEST;
  - (iii) The three sets of best-fit parameters (Table 1) were used again to generate 100 stochastic realizations (rainfall time series), for the wet, normal and dry year, respectively. These realizations may deviate from the actual rainfall data in terms of daily rainfall values, still being statistically identical with the rainfall pattern of the representative years.

Table 1. Three parameter sets of the rainfall generator for the wet, normal and dry year,respectively.

| Tespectively.                            |                   |                    |                   |                  |                                                    |  |  |
|------------------------------------------|-------------------|--------------------|-------------------|------------------|----------------------------------------------------|--|--|
| Parameter                                | Wet Year (2007)   | Normal Year (2008) | Dry Year (2018)   | Adjustable range | Note                                               |  |  |
| P(mm)                                    | 916.3             | 588.7              | 444.1             | (400,1000)       | Annual precipitation                               |  |  |
| $P_{drizzle}$                            | 0.137             | 0.146              | 0.140             | (0.01, 0.99)     | Probability of drizzle events                      |  |  |
| Monthly mean storm duration γ            |                   |                    |                   |                  |                                                    |  |  |
| $\gamma_s(\mathrm{day})$                 | 2.546             | 2.177              | 2.071             | (0.1, 10)        | Seasonally averaged storm duration                 |  |  |
| $\alpha_{\gamma}(\mathrm{day})$          | 1.764             | 0.820              | 0.550             | (-2.2, 2.2)      | Amplitude of seasonal storm shift                  |  |  |
| $	au_{\gamma}(\mathrm{day})$             | 105.760           | 100.297            | 100.474           | /                | Phase of seasonal storm shifts                     |  |  |
| Monthly mean inter-storm period $\delta$ |                   |                    |                   |                  |                                                    |  |  |
| $\delta_s(\mathrm{day})$                 | 7.988             | 7.424              | 7.535             | (0.1, 10)        | Seasonally averaged inter-storm period             |  |  |
| $\alpha_{\delta}(\mathrm{day})$          | 2.457             | 2.962              | 2.946             | (-8, 8)          | Amplitude of seasonal inter-storm shift            |  |  |
| $	au_{\delta}(\mathrm{day})$             | 120.665           | 124.044            | 124.161           | /                | Phase of seasonal inter-storm shifts               |  |  |
| Expected storm intensity $E[i t_r]$      |                   |                    |                   |                  |                                                    |  |  |
| $(a_1, b_1)^1$                           | (0.875,           | (0.934,            | (0.909,           | (0.1, 5)         | Coefficient 1 and 2 of the                         |  |  |
| $(u_1, b_1)$                             | 2.349)            | 1.802)             | 1.598)            |                  | first season (Jan-Mar)                             |  |  |
| $(a_1,b_1)^2$                            | (1.015,<br>0.661) | (0.894,<br>0.538)  | (0.857,<br>0.456) | (0.1, 5)         | Coefficient 1 and 2 of the second season (Apr-Jun) |  |  |
| $(a_1, b_1)^3$                           | ŕ                 | (0.968,            | ŕ                 | (0.1, 5)         | Coefficient 1 and 2 of the                         |  |  |
|                                          | (0.917,<br>1.121) | 0.845)             | (0.947,<br>0.752) |                  | Coefficient 1 and 2 of the third season (Jul-Sep)  |  |  |
| $(a_1, b_1)^4$                           | (1.666,           | (1.348,            | (1.392,           | (0.1, 5)         | Coefficient 1 and 2 of the                         |  |  |
|                                          | 1.556)            | 1.341)             | 1.263)            |                  | fourth season (Oct-Dec)                            |  |  |
| expected storm variability $CV^2[i t_r]$ |                   |                    |                   |                  |                                                    |  |  |
| (a h)                                    | (0.658,           | (0.617,            | (0.632,           | (0.1, 5)         | Coefficient 1 and 2                                |  |  |
| $(a_2, b_2)$                             | 0.961)            | 1.057)             | 1.025)            |                  |                                                    |  |  |

#### 3.4 Simulation scenarios

#### 3.4.1 Inter-annual rainfall variability

Wet Year (WY), Normal Year (NY), and Dry Year (DY) (Table 1) were set as the simulation scenarios. In addition, a fourth scenario consistent with the Dry Year but coupled with reduced plant uptake was considered as Extreme Dry Year (EDY). This EDY scenario was used to account for the extreme drought occurring in 2018 that caused vegetation to die back and impacted N dynamics to some extent [Winter et al., 2023]. In the EDY scenario, the plant uptake was assumed to decrease down to 36% of the original value used in the DY scenario, according to the classification and the occurrence period of the drought [Liu et al., 2010]. For each of the scenarios, the water flow and N transport model were conducted 100 times by substituting the rainfall data during the simulation period with the generated 100 stochastic rainfall time series (Figure S1, Supporting Information). Average annual N loads and fluxes in each scenario were calculated to ensure that they were not controlled by a single realization and statistically meaningful. Additionally, we simulated the fluctuations of C<sub>Q</sub> in each scenario with time over three years. Finally, the average annual N loads, fluxes, and C<sub>Q</sub> as well as the variation of C<sub>Q</sub> were cross-compared and analyzed to elucidate the effect of inter-annual rainfall variability on N dynamics.

#### 3.4.2 Intra-annual rainfall variability

In order to explore the effect of intra-annual rainfall variability on N dynamics, the

linear regression analyses between the parameters of the stochastic rainfall generator and N loads, fluxes as well as CQ were conducted, in which the parameters of NY serve as a reference. The seasonal averages of storm duration and inter-storm period  $(\gamma_s$  and  $\delta_s$ ), the amplitudes of the seasonal variations in storm duration and inter-storm period ( $\alpha_{\gamma}$  and  $\alpha_{\delta}$ ), the average rainfall intensity of four seasons (E<sub>1</sub>~E<sub>4</sub>) and squared coefficient of variation of the average rainfall intensity (CV<sup>2</sup>), as well as the probability of drizzle events (Pdrizzle), were selected as the experimental parameters.  $E_1 \sim E_4$  and  $CV^2$  can be approximately calculated by equation S5 and S6, using relative parameters and seasonally averaged storm duration  $(\gamma_s)$ . In total, 10 parameters (Table 1) were subjected to the assessment of their effect on N dynamics. For each of the experimental parameters, the rainfall generator generated 100 rainfall time series randomly, whose annual precipitations are within the historical range, with this experimental parameter varying randomly within the adjustable range but other parameters being fixed to the best-fit values for the NY scenario. The ranges of the experimental parameters used in the linear regression analyses are listed in Table 1. Finally, the response of the annual N loads, fluxes and Co to different rainfall parameters can be analyzed.

## 4 Results

The calibrated N transport model showed good performance in fitting the in-stream nitrate concentration (Figure 4a), with a Nash-Sutcliffe efficiency (NSE) of 0.79. The

modeled N surplus of 51.87 kg ha<sup>-1</sup> yr<sup>-1</sup> is comparable to the measured value of 48.8 412 kg ha<sup>-1</sup> yr<sup>-1</sup>. The calibrated best-fit values for the transport parameters are listed in 413 Table S1 (see Supporting Information). 414 415 Figure 3 illustrates the 14-year N mass balance simulated by the calibrated N 416 transport model in the entire catchment. In the soil source zone, the total N consisted of SON (552 kg ha<sup>-1</sup>, including SON(a) of 90 kg ha<sup>-1</sup> and SON(p) of 462 kg ha<sup>-1</sup>) and 417 SIN (48 kg ha<sup>-1</sup>). The load of SON accounts for 92% of the total N, which 418 419 corresponds to the research result that the organic N fraction is greater than 90% 420 [Stevenson., 1995]. As for N transformation, the mineralization rate of 173 kg ha<sup>-1</sup> yr<sup>-1</sup> is within the range (14-187 kg ha<sup>-1</sup> yr<sup>-1</sup>) reported by Heumann et al. [2011] for study 421 422 sites located in Germany. 71% of the SIN was absorbed by plants at a rate of 123 kg ha<sup>-1</sup> yr<sup>-1</sup>, which is very close to the value (around 120 kg ha<sup>-1</sup> yr<sup>-1</sup>) suggested in 423 Nguyen et al. [2021] for the same area. The denitrification flux of 38 kg ha<sup>-1</sup> yr<sup>-1</sup> (4 kg 424 ha<sup>-1</sup> yr<sup>-1</sup> in the soil source zone, 34 kg ha<sup>-1</sup> yr<sup>-1</sup> in the groundwater) is within the range 425 (8-51 kg ha<sup>-1</sup> yr<sup>-1</sup>) investigated for 336 agricultural areas around the world by Hofstra 426 and Bouwman [2005]. The SIN entered the groundwater zone at a rate of 44 kg ha<sup>-1</sup> 427 yr<sup>-1</sup>, which is within the range (15-60 kg N ha<sup>-1</sup> yr<sup>-1</sup>) reported in Nguyen et al. [2021]. 428 Approximately 23% of the leachate of SIN was exported to the stream, where the rest 429 430 of the leachate (77%) was removed by denitrification. The export and denitrification 431 fluxes are comparable to the values 39% and 61% reported in Nguyen et al. [2021]. 432 Based on measured groundwater chemistry data from nearby catchments and multiple estimates of denitrification potential that have been proved and reported [Hannappel et al., 2018], we conclude that the simulated transformation and transport of N are acceptable.

Figure 3. The simulated 14-year N mass balance in the entire catchment.

## 4.1 Effect of Inter-annual rainfall variability

**Figure 4.** The fluctuations of simulated in-stream nitrate concentration (C<sub>Q</sub>) under (a) the historical rainfall (1997-2007) and the scenarios of (b<sub>1</sub>) WY, (b<sub>2</sub>) NY, (b<sub>3</sub>) DY, and (b<sub>4</sub>) EDY (2008-2010). The grey areas are formed by the realizations of each scenario.

**Figure 5.** The simulated annual in-stream nitrate concentration ( $C_Q$ ) for scenarios WY, NY, DY, and EDY. The whisker represents the concentration ranges from the 5<sup>th</sup> to 95<sup>th</sup> percentiles, with the triangles indicating the median and the lines marking the maximum, average, and minimum of the data set from top to bottom. The same as below.

The variations of  $C_Q$  in these scenarios are shown in Figure  $4b_1 \sim 4b_4$ . The range of  $C_Q$  generally decreases when the rainfall pattern transforms from WY, via NY, to DY, but the range of EDY is wider than that of WY. Figure 5 shows the range of the simulated annual in-stream nitrate concentration for the four scenarios. WY produced higher  $C_Q$ , with the median concentration reaching nearly 5 mg  $L^{-1}$ . The concentrations generally decreases when the rainfall pattern transforms from WY, via NY, to DY. However, when vegetation dieback occurred, most concentration values in the EDY scenario were far higher than those of WY. As shown in the results, the median concentration of EDY reached 8 mg  $L^{-1}$ . The soil organic nitrogen (SON(a) and SON(p)) loads increase when the rainfall pattern shifts from WY, via NY, to

DY/EDY (Figure 6a and 6b). The SON loads in DY and EDY were identical because the transformation of SON is not subject to vegetation state. There were no remarkable differences in soil inorganic nitrogen (SIN) load among WY, NY, and DY (Figure 6c). However, the SIN load in EDY was the highest among the four scenarios.

**Figure 6.** The simulated loads of (a) active soil organic nitrogen (SON(a)), (b) protected soil organic nitrogen (SON(p)), and (c) soil inorganic nitrogen (SIN) for scenarios WY, NY, DY, and EDY.

The highest and the lowest average mineralization levels were produced in WY and DY, respectively (Figure 7a). DY and EDY had identical mineralization fluxes. Note that the fluxes of plant uptake, denitrification (in soil and groundwater), and leaching are negative values representing sink terms for the SIN pool (Figure 7b, 7c and 7d). These fluxes decrease when the rainfall pattern transforms from WY, via NY, to DY, due to decreasing soil moisture. In EDY, vegetation dieback resulted in the lowest level in plant uptake. Although the soil moisture content in EDY was low, denitrification and leaching fluxes were larger than those of WY. The N export flux to the stream generally follows the same trends as leaching, with relatively higher fluxes

occurring in WY and EDY and lower fluxes in NY and DY (Figure 7e). Based on these results, it can be preliminarily concluded that annual precipitation and vegetation state critically influence N transformation and transport.

**Figure 7.** The simulated fluxes of (a) mineralization, (b) plant uptake, (c) denitrification (in soil and groundwater), (d) leaching, and (e) N export for scenarios WY, NY, DY, and EDY.

## 4.2 Effect of Intra-annual rainfall variability

The determination coefficients (R<sup>2</sup>) of the linear regressions between the parameters of the rainfall generator and the  $C_Q$ , N loads, and fluxes are illustrated in Figure 8a. Larger R<sup>2</sup> values can be observed for the amplitude of the seasonal variations in the storm duration and inter-storm period ( $\alpha_{\gamma}$  and  $\alpha_{\delta}$ ), the average rainfall intensity of

the third season (E<sub>3</sub>) as well as the probability of drizzle events (P<sub>drizzle</sub>). These parameters are the four most important parameters (Figure 8b) that influence N dynamics, and are subject to further discussion.

**Figure 8.** (a) The determination coefficients  $(R^2)$  of the linear regressions between the parameters of the rainfall generator and in-stream nitrate concentration  $(C_Q)$ , N loads, and fluxes. (b) The importance of the parameters influencing the N dynamics.

#### Storm duration

The parameter  $\alpha_{\gamma}$  determines how the monthly average storm duration is distributed over the course of a year (Figure 9a). Generally, larger  $\alpha_{\gamma}$  values represent that storms with longer duration are more likely to occur in mid-year when ET is high and the inter-storm duration is longer (yellow zone, Figure 9e). This pattern not only influences the drying and rewetting cycles in summer, but also supplies more water during this period, thereby leading to an overall wet year (e.g., year 2007,  $\alpha_{\gamma}$ =1.764). A decrease in  $\alpha_{\gamma}$  causes storms with longer durations to shift towards the beginning/end of the year, when ET is low and the inter-storm duration is shorter

(blue zone, Figure 9e), so that the year becomes drier (e.g., year 2018,  $\alpha_v = 0.55$ ).

Results suggest that the storm duration distribution during a year significantly affects N transformation and transport. Lower SON loads (Figure 9a) and higher SIN loads (Figure 9b) occurred when longer-duration storms concentrate in mid-year. This is because wetter catchment conditions promote the mineralization of SON into SIN, especially during warm periods when mineralization is primarily constrained by soil moisture rather than temperature. The wetter conditions associated with larger  $\alpha_{\gamma}$  values increased the leaching flux of SIN from soil to groundwater (Figure 9c), which elevated the average  $C_Q$  (Figure 9d). When shorter-duration storms dominate mid-year, N transformation and transport were inhibited.

**Figure 9.** The effect of varying monthly average storm duration distributions over the course of a year on (a) SON load, (b) SIN load, (c) leaching flux (LEA), and (d) in-stream nitrate concentration ( $C_Q$ ). (e) The monthly average inter-storm period distribution over the course of a year ( $\alpha_\delta$  is kept constant when  $\alpha_\gamma$  varies). As  $\alpha_\gamma$  values decrease, longer-durations storms shift gradually from the period with high ET and longer inter-storm periods (yellow zone) to the period with low ET and shorter inter-storm periods (blue zone).

#### Inter-storm period

 $\alpha_{\delta}$  regulates the variation in monthly average inter-storm periods throughout the

course of a year (Figure 10a). Larger  $\alpha_{\delta}$  values cause longer inter-storm periods to cluster primarily in mid-year, coinciding with longer storm durations (blue zone, Figure 10e), while shorter inter-storm periods concentrate at the beginning/end of the year, associated with shorter storm durations (yellow zone, Figure 10e). This forms a "wet-dry-wet" climate pattern that promoted SON accumulation (Figure 10a) but reduced SIN load (Figure 10b), leaching flux (Figure 10c), and  $C_Q$  (Figure 10d). Conversely, decreasing  $\alpha_{\delta}$  redistributes longer inter-storm periods to the year's boundaries and shorter inter-storm periods to mid-year, thereby forming a "dry-wet-dry" climate pattern over a year. It resulted in a lower SON load and a higher SIN load, as mid-year warmth and moisture promote the transformation and transport of SON accumulated in the previous season. Consequently, leaching flux and  $C_Q$  significantly increased.

**Figure 10.** The effect of varying monthly average inter-storm period distributions over the course of a year on (a) SON load, (b) SIN load, (c) leaching flux (LEA), and (d) in-stream nitrate concentration ( $C_Q$ ). (e) The monthly average storm duration distribution over the course of a year ( $\alpha_\gamma$  is kept constant when  $\alpha_\delta$  varies). As  $\alpha_\delta$  values decrease, longer-periods inter-storms shift gradually from the period with longer storm durations (blue zone) to the period with shorter storm durations (yellow zone).

#### Average Storm Intensity

Figure 11 shows the responses of N loads, fluxes, and in-stream nitrate concentration

(C<sub>0</sub>) to seasonal average rainfall intensity (E<sub>1</sub>-E<sub>4</sub>). Fundamentally, increased rainfall intensity creates more humid conditions that intensify biogeochemical processes (e.g., mineralization, plant uptake). Based on linear regression slopes (k), E2 and E3 generally exert stronger effects on N loads and fluxes than E1 and E4. Because mineralization that is the only source of SIN remained unconstrained by temperature during the warm seasons (Figure 11b4 and 11c4). Elevated E2/E3 enhanced the transformation of SON(a) (Figure 11b<sub>1</sub> and 11c<sub>1</sub>) and SON(p) (Figure 11b<sub>2</sub> and 11c<sub>2</sub>), whose loads significantly reduced. Although SIN load theoretically should increase with rainfall intensity, as the only SON sink, it decreased in the second season (Figure 11b<sub>3</sub>) due to substantially increased plant uptake (Figure 11b<sub>5</sub>). Warmer temperatures and enhanced moisture during the growing season promote vigorous nutrient absorption by vegetation. In addition, soil denitrification rose markedly during the third season as average rainfall intensity increased (Figure 11c<sub>6</sub>), which is due to favorable microbial conditions and elevated SIN loads. In warm periods, enhanced average rainfall intensity increased actual evapotranspiration (ET) (Figure S2b<sub>1</sub> and S2c<sub>1</sub>), which correspondingly should reduce recharge and discharge. However, both recharge and discharge increased during the second season (Figure S2b<sub>2</sub> and S2b<sub>3</sub>). Export flux and C<sub>0</sub> reduced slightly in the second season (Figure S2b<sub>4</sub> and 11b<sub>7</sub>) but elevated in the third season (Figure S2c<sub>4</sub> and 11c<sub>7</sub>), mirroring SIN load trends.

**Figure 11.** The responses of N loads, mineralization, plant uptake, and denitrification (in soil) fluxes, as well as in-stream nitrate concentration ( $C_Q$ ) to the average rainfall intensity of the four seasons ( $E_1$ - $E_4$ ). The sign and magnitude of the slopes (k) in these linear relationships

denote the direction and the intensity of response of N dynamics to the variations in average rainfall intensity, respectively.

#### Probability of drizzle event

Figure 12 illustrates the responses of N loads, fluxes, discharge (Q) and in-stream nitrate concentration ( $C_Q$ ) to the probability of drizzle events ( $P_{drizzle}$ ).  $P_{drizzle}$  characterizes the pattern of precipitation intensity over the course of a year. A low  $P_{drizzle}$  value indicates a predominance of moderately intense rainfall evenly distributed throughout the year, resulting in consistently humid conditions (e.g., year 2007,  $P_{drizzle}$ =0.137). Conversely, a high  $P_{drizzle}$  value represents the situation with high-frequency drizzle and low-frequency extreme precipitation, forming drought-flood extremes. With the increase of  $P_{drizzle}$  (climate transition from consistently humid conditions to extreme drought-flood patterns), SON load decreased due to the enhanced transformation (Figure 12a<sub>1</sub> and 12b<sub>1</sub>). SIN load decreased (Figure 12c<sub>1</sub>) due to the elevated plant uptake (Figure 12a<sub>2</sub>). Denitrification and leaching fluxes were reduced (Figure 12b<sub>2</sub> and 12c<sub>2</sub>), while  $C_Q$  declined with increasing  $P_{drizzle}$  (Figure 12c<sub>3</sub>).

**Figure 12.** The responses of N loads, fluxes, discharge (Q) and in-stream nitrate concentration ( $C_Q$ ) to the probability of drizzle events ( $P_{drizzle}$ ). The sign and magnitude of the slopes (k) in these linear relationships denote the direction and the intensity of response of N dynamics to the variations in average rainfall intensity, respectively.

# 5 Discussion

## 5.1 N transformation and transport upgrade in wet years

The comparison of C<sub>Q</sub>, N loads, and fluxes across four scenarios (WY, NY, DY, and EDY) reveals the effect of inter-annual rainfall variability on N transformation and

transport, as well as water quality. Mineralization, the crucial process of the transformation from SON to SIN, exhibits strong soil moisture dependence. Consequently, the highest average mineralization rates in the WY scenario promoted N transformation through plant uptake, denitrification, and leaching. In contrast, low mineralization in the DY scenario led to SON accumulation, thereby restraining overall N dynamics. Additionally, the transport of SIN relies on groundwater discharge and subsurface flow paths. Increased discharge from precipitation changes flushes more nitrogen into surface water bodies [Mitchell et al., 1996; Creed and Band, 1998] via fast shallow flow paths with short transit times (I-I cross section, Figure 1a) [Yang et al., 2018]. Consequently, N export and C<sub>O</sub> variations align with wetness changes among WY, NY, and DY. These results demonstrate that high annual precipitation enhances N transformation and transport, while low annual precipitation promotes catchment N retention. Our simulation results echo previous findings that N retention is influenced by changes in precipitation [Dumont et al., 2005; Howarth et al., 2006].

## 5.2 Stream water quality deteriorates during extreme droughts

SIN significantly accumulated in soil when the vegetation suffered from drought stress [RIVM, 2021; Winter et al., 2023]. This is because in general plant uptake is the major sink for SIN [Yang et al., 2021; Zhou et al., 2022; Wang et al., 2023]. Due to the accumulation of SIN, denitrification and leaching in the EDY scenario were

higher than those in the DY scenario. Thus, N export to the stream in the EDY scenario was higher than that in the DY scenario, even though the stream discharge was the same (Figure 13). This resulted in higher  $C_Q$  in the EDY scenario than that in the WY scenario.

Figure 13. The simulated discharge for scenarios WY, NY, DY, and EDY.

During the 2018-2019 drought in central Europe, the observed peaks of in-stream nitrate concentration at a mesoscale catchment in central Germany were significantly higher than the previous concentration peaks (Figure 14). Based on the data-driven analysis, Winter et al. [2023] indicated that nitrate loads in the 2018-2019 drought were up to 73% higher than the long-term average loads. They demonstrated that such increases were attributed to decreased plant uptake and subsequent flushing of accumulated nitrogen during the rewetting period. Our results confirm their findings that reduced plant uptake in extreme droughts causes C<sub>Q</sub> to elevate significantly (Figure 4b<sub>4</sub>) and the N retention capacity to decline. These results suggest that vegetation state plays a vital role in the increased risk of N pollution during extreme

#### droughts.

Figure 14. Measured daily averaged nitrate concentrations (C, red line) and discharge (Q, black line) in the nearby Upper Selke catchment in central Germany [Winter et al., 2023]. The yellow area marks the 2018-2019 drought occurring across central Europe.

#### 5.3 The distribution of storm durations controls the N loads

Our study indicates that the wet/dry conditions in mid-year are key to N transformation and transport throughout the whole year. The combination of longer storm durations and extended inter-storm periods in mid-year represents an aspect of extreme climate events where intense rainfalls alternate with droughts [Zhang et al., 2024]. Without temperature limitations, alternating longer storm durations and longer inter-storm periods increase the extensive SON transformation, which is consistent with the results of laboratory studies that mineralization increased after successive drying and rewetting cycles in soils [Cabrera, 1993]. It is concluded that drying-rewetting events induce the prominent changes in N dynamics [Fierer and

Schimel, 2002]. However, the drier conditions induced by shortened storm durations and longer inter-storm periods in mid-year cause the accumulation of SON, which is consistent with the finding by Zhou et al. [2022] that the drought in the warm season contributed to catchment nitrogen retention. Thus, storm duration distributions within a year can significantly affect N loads in a catchment.

## 5.4 The distribution of inter-storm periods changes stream water

### quality

Different dry-wet patterns over the course of a year have an impact on N dynamics, particularly in-stream nitrate concentration. In the "dry-wet-dry" pattern, accumulated SON from the dry season is transformed and transported during the subsequent wetting period, which leads to higher C<sub>Q</sub>. During the 2018-2019 drought, accumulated N loads were flushed during the rewetting period and thereby caused elevated nitrate loads (Figure 14). In the "wet-dry-wet" pattern, humid conditions with low temperatures in the year's boundaries and warm droughts in mid-year both cause SON loads to accumulate throughout the entire year and C<sub>Q</sub> to decrease. Zhou et al. [2022] attributed reduced C<sub>Q</sub> to limited terrestrial export loads. In summary, inter-storm period distributions cause noticeable variations in water quality in terms of nitrate concentration.

# 5.5 Average rainfall intensity has only a small effect on stream water

## quality

Wet conditions caused by high mean rainfall intensities enhance the transformation of SON to SIN [Knapp and Smith, 2001], when mineralization is not constrained by low temperatures (Equation 1 and 2). The potential plant uptake is subject to vegetation state (Equation 6), while the actual plant uptake is further limited by SIN load and soil moisture content (Equation 5) [Yang et al., 2022; Cramer et al., 2009]. Thus, high mean rainfall intensities promote plants absorption of SIN during active growth periods. Since plant uptake is the major sink of SIN, the responses of SIN load to mean rainfall intensity variations are jointly determined by mineralization and plant uptake, resulting in complex SIN load dynamics. SIN load controls denitrification in soil and leaching. However, leaching is a complex process additionally influenced by soil saturation and groundwater velocity (Equation 3 and 7). As a result, no significant linear relationship exists between mean rainfall intensity and leaching. Both soil denitrification flux and N export flux (a component of leaching flux) vary with the average rainfall intensity, following SIN load patterns. It is noteworthy that high-intensity precipitation events with short durations and substantial surface runoff rarely propagate to the groundwater zone, minimally affecting discharge and nitrate export (Figure S2, Supporting Information). Therefore, Co has weaker responses in extreme precipitations.

## 5.6 Probability of drizzle event alters N dynamics

Extreme dry-wet patterns enhance the transformation of SON more effectively than continuously humid conditions, which echoes laboratory studies conducted by Cabrera [1993] that successive drying and rewetting cycles in soils can intensify mineralization. As the major sink, elevated plant uptake due to increased SIN load in turn reduced SIN load, which resulted in weaker denitrification and leaching. This is because denitrification and leaching are controlled by SIN load (Equation 3, 4, and 7). Following the same trends as leaching, N export flux slightly decreased. More discharge yielded in extreme dry-wet patterns than in continuously humid conditions. The combination of increased discharge and slightly reduced N export led to a decrease in C<sub>0</sub>. It follows that the probability of drizzle events prominently alters N dynamics. On the whole, increased variations in precipitation can alter the transformation and transport of nitrogen [Kane et al., 2008]. The results of the present study can help in managing the water quality of agricultural catchments with prominent rainfall variability and protecting aquatic ecosystems during extreme rainfall and droughts. For instance, a fertilization scheme for inter-storm periods can be formulated to avoid flushing during the rewetting periods after droughts. Also, increased fertilization in the growing season with humid conditions is not only conducive to vegetation absorption of nutrients, but also does not cause pressure on soil and water bodies.

### 5.7 Limitations and Outlook

The transport model preserves the main pathway for N-NO<sub>3</sub>-leachate by simplifying complexities of different N pools and transformations via mineralization, leaching, and denitrification within the soil zone. However, the external N input entering directly into inorganic N, the transformation from protected organic N to active organic N, and the loss of organic N via dissolution have not been included in the model, which may lead to miscalculation of nitrogen load. As a complex biogeochemical process, denitrification is governed by various factors such as temperature, soil moisture content, and SIN load, rather than a first-order decay process. Nonetheless, the current transport model does not account for spatial heterogeneity in denitrification. These simplifications may introduce uncertainties in the simulated results. In addition, due to the lack of groundwater nitrate concentration data, the nitrate transport model was calibrated using only in-stream nitrate concentration data [Wang et al., 2023]. More nitrate concentration data in groundwater and river will contribute to establishing a more realistic nitrate transport model. The plant-soil process was not represented in the model. In the extreme dry scenario, plant death was not actually simulated under high temperature and low soil moisture content, but was manually assigned a lower plant uptake potential. The plant growth stages were assumed to be constant. This is because the N uptake process was only

mathematically described using the empirical formula (Equation 6) in the simplified N framework, rather than using a full plant-soil process. However, such simplification was proved to be effective in terms of reproducing reasonable N loads, fluxes and in-stream nitrate concentrations. Therefore, we think it is acceptable to identify the key effect of plant die-off (i.e. reduced N uptake potential) on stream water quality during extreme dry scenarios. In the future, the alteration of soil properties caused by rainfall variability should be studied. In the model, precipitation and solute enter the subsurface domain with fixed soil porosity, ignoring the alteration of pores. Abundant microorganisms and animals engage in extensive activities, and massive organic and inorganic matter undergoes biochemical reactions in soil pores, which all change the physical and chemical properties of porous media and thereby impact seepage. These biogeochemical processes are impacted by several factors, such as hydrometeorological conditions [Ondrasek et al., 2019]. For example, the activities of microorganisms and animals degrade under drying conditions, which is not beneficial for maintaining and improving soil structures. The severe deficit of soil moisture results in the disintegration of soil aggregates, and even compacted soil forming in the upper layer due to salt clusters, which ultimately reduces porosity. Additionally, cracked surface soil caused by extreme droughts is detrimental to plant growth and root development. Therefore, altered soil properties induced by complex and extreme climate patterns can substantially influence soil biogeochemical activities. The clarification of these

mechanisms will be conducive to exploring the more realistic response of solute transport in groundwater to extreme rainfalls and droughts.

# 6. Conclusions

- Within the context of climate change and increasing occurrence of extreme rainfalls and droughts, the study pioneered the application of a stochastic rainfall generator (Robinson and Sivapalan, 1997) coupled with the flow and transport model to systematically investigate inter-annual and intra-annual rainfall variability effects on N dynamics in a small agricultural catchment located in Central Germany. The principal findings are as follows:
- (1) Higher annual precipitation enhances N transformation and transport, whereas

  lower annual precipitation is conducive to N retention capacity. Nonetheless, the

  retention capacity declines severely when vegetation suffers from drought stress.

  Vegetation plays a vital role in N dynamics especially during extreme droughts.
- (2) Wet/dry conditions determined by storm duration distributions within a year can
  significantly affect N loads in a catchment. Droughts can prompt the accumulation
  of SON, but a drying-wetting cycle can enhance extensive SON transformation in
  the warm season.
- (3) Different dry-wet patterns formed by inter-storm period distributions during a year

  can cause noticeable variations in in-stream nitrate concentrations that

- prominently elevate during the rewetting period after drought.
- (4) Elevated mean rainfall intensity contributes to N transformation when
- mineralization is not constrained by low temperatures, and promotes plant
- absorption of SIN during the growing season. There is merely a small effect on
- stream water quality.
- (5) The probability of drizzle events, characterizing the pattern of precipitation
- intensity within a year, prominently alters N dynamics. Extreme dry-wet patterns
- enhance SON transformation and plant uptake but inhibit denitrification and
- leaching more effectively than continuously humid conditions.
- Overall, the study clarifies the effect of rainfall variability on N dynamics in a small
- agricultural catchment, which provides theoretical support for formulating
- fertilization strategies and protecting aquatic ecosystems in the context of climate
- change.
- *Code availability*. All data used in this study are listed in the supporting information
- and uploaded separately to HydroShare [Wang, 2024].
- Author contributions. QW contributed to the conceptualization, methodology,
- software, formal analysis, visualization, and writing (review and editing). JY
- contributed to the conceptualization, methodology, formal analysis, and writing
- (review and editing). IH contributed to the writing (review and editing). TX
- contributed to the conceptualization and review and editing. CL contributed to the

- methodology, and review and editing.
- *Competing interests*. The contact author has declared that none of the authors has any
- competing interests.
- Disclaimer. Publisher' note: Copernicus Publications remains neutral with regard to
- jurisdictional claims in published maps and institutional affiliations.
- Acknowledgements. We thank Daniel C. Wilusz for the source code in Python v2.7 of
- the rainfall generator [Wilusz et al., 2017] and Min Yan and Huiqiang Wu for
- stimulating discussions.
- Financial support. This research was supported by the National Key Research and
- Development Project, China (JY: 2024YFC3211600), the National Natural Science
- Foundation of China, China (JY & CL: U2340212, TX: 42377046, CL: 51879088),
- the Fundamental Research Funds for the Central Universities, China (JY:
- B250201002), the German Research Foundation DFG, Germany (IH: 454619223),
- and the Natural Science Foundation of Jiangsu Province, China (CL: BK20190023).

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
