# Peer review of "The effect of rainfall variability on Nitrogen dynamics"

_EGUsphere, 2025_

## Author Comment (AC3)

Dear reviewer,

Thanks you so much for your constructive comments for the discussion phase. Please check our reply for all your comments point by point (followed and marked in blue). However, the revised completed manuscript will be provided in the next step of the review processes.

Best regards

**Responses:**

This manuscript presents a numerical modeling study exploring how inter- and intra-annual precipitation variability affects nitrogen (N) loads and fluxes in a catchment. The topic is timely and important for water quality and environmental management. The paper is generally well written, though some grammatical edits are needed. The methods and results are mostly clear and logically presented. The conclusions are relevant and likely to be of interest to scientists and resource managers focused on mitigating nutrient pollution.

Primary suggestions are to:

(1) Clarify and narrow the research focus: While the broad importance of precipitation variability is well established in the introduction, the final introductory paragraph should better define the unique contribution this work makes to the literature. Consider making the statement of objectives more specific and clarifying how the objectives fill gaps left by other recent studies.

Response #1:

Thanks for point that out! We have rephrased the last paragraph of the introduction part to define the unique contribution this work as " *In order to fill the gap, the present study explored the impact of rainfall variability on N dynamics and its potentially negative influence on water quality across inter-annual and intra-annual timescales. To characterize rainfall variability, a stochastic rainfall generator [Robinson & Sivapalan, 1997] was employed to generate rainfall time series with different climatic characteristics. The research was conducted in a small agricultural catchment located in Central Germany, where a hydrological model was previously established utilizing*

*the fully coupled surface-subsurface numerical simulator HydroGeoSphere [Yang et al., 2018]. The framework of N dynamics was modified from the ELEMeNT approach [Exploration of Long-tErM Nutrient Trajectories; Van Meter et al., 2017]. The research is divided into two main components. First, three representative years (with high, normal, and low annual precipitation amounts, respectively) were chosen from the past two decades in Central Germany as the target scenarios. A fourth scenario with low annual precipitation amounts coupled with reduced plant uptake represents a case where vegetation is partially destroyed by extreme droughts. The statistical analyses of nitrogen loads and fluxes and a comparison across different scenarios were conducted to reveal the influence of inter-annual rainfall variability. Second, rainfall time series generated using the stochastic rainfall generator by separately altering specific parameters were used to substitute the rainfall data in the simulation period to drive the flow and nitrogen transport models. The responses of the N loads and fluxes to the parameters (e.g., the amplitudes of the seasonal variations in the storm durations and inter-storm period) were thoroughly analyzed to clarify the effect of intra-annual rainfall variability. The study will provide theoretical support to formulate fertilization strategies and protect aquatic ecosystems under climate change in the future.*"

Reference:

*Robinson, J., & Sivapalan, M.: Temporal scales and hydrological regimes: Implications for flood frequency scaling, Water Resour. Res., 33(12): 2981-2999, https://doi.org/10.1029/97WR01964, 1997.*

*Yang J., Heidbüchel, I., Musolff, A., Reinstorf, F., and & Fleckenstein. J. H.: Exploring the dynamics of transit times and subsurface mixing in a small agricultural catchment, Water Resour. Res., 54, https://doi.org/ 10.1002/2017WR021896, 2018.*

*Van Meter, K. J., Basu, N. B., Van Cappellen, P.: Two centuries of nitrogen dynamics: Legacy sources and sinks in the Mississippi and susquehanna river basins, Global Biogeochem. Cycles, 31 (1), 2–23., https://doi.org/10.1002/2016GB005498, 2017.*

(2) Differentiate from previous work: The study references and builds on previous work, particularly Wang et al. (2023). However, the manuscript does not always make it clear

where prior work ends and the current study begins. The manuscript also refers the reader to the previous studies for some crucial details (e.g. calibration to stream nitrate concentrations), which are not easy to find in the previous studies. I recommend adding further descriptions of the related studies at this site and delineating which analyses, model developments, and findings are new in this study.

Response#2:

Thanks for the points!

Generally, this study employed the model developed in our previous work Wang et al. [2023] for a same study site. The model setup, method for simulating the flow and N transport, parameters and the calibration are all from Wang et al. [2023]. However, new contributions are attributed to exploring the effect of rainfall variability on N dynamics. To clarify the existing work and the new contributions, we rephased the method section such that the new contributions are highlighted. Additionally, more details regarding the model calibration were added in the results section as a short review: "*The calibrated N transport model showed good performance in fitting the in-stream nitrate concentration (Figure 3a), with a Nash-Sutcliffe efficiency (NSE) value of 0.70. Subsequently, the calibrated model was validated over the entire simulation period from 1997 to 2010 (~1 day of CPU time), with a NSE value of 0.55. Due to incomplete observation and the high CPU time, the NSE is still considered to be acceptable. The calibrated best-fit values for the transport parameters are listed in Table S1 [see in Supporting Information]. More detailed calibration results refer to Wang et al. [2023].*"

(3) Clarify model-data connection: A clearer connection could be made between the numerical model and real-world observations. The use of simple abstractions in the nutrient transport component of the model is reasonable, but also requires careful consideration of (a) how well the simplified processes representations mimic actual processes, (b) the uncertainty of the parameter estimates, and (c) the accuracy of the model in terms of reproducing observations. Otherwise, there is a risk of circularity: the model is built around certain processes and parameterizations, and then used to test the importance of those same processes and parameters.

Response#3:

Thanks for the suggestions. We made the following changes to clarify model-data connection:

1.  We added the observed in-stream nitrate concentrations along with the simulated ones (see the figure below) such that they can be clearly compared.

[Figure]

2.  We added a description of the calibration results to show the accuracy of the model in terms of reproducing observations as "The calibrated N transport model showed good performance in fitting the in-stream nitrate concentration (Figure 3a), with a Nash-Sutcliffe efficiency (NSE) value of 0.70. Subsequently, the calibrated model was validated over the entire simulation period from 1997 to 2010 (~1 day of CPU time), with a NSE value of 0.55. Due to incomplete observation and the high CPU time, the NSE is still considered to be acceptable. The calibrated best-fit values for the transport parameters are listed in Table S1 [see in Supporting Information]. More detailed calibration results refer to Wang et al. [2023]".

3.  We added short discussion about the uncertainty of the parameter in the section of Discussion.

Line-by-Line comments:

(4) Line 14: "performance" is wrong word. "effect"?

Response#4:

Thanks! We changed "performance" into "manifestation".

(5) 34: SON not defined at this point

Response#5:

Thanks, we rephrased as "..SON (soil organic nitrogen)…".

(6) 39 – "a small effect"

Response#6:

We corrected accordingly.

(7) 58 – not clear what "their" refers to

Response#7:

We clarified it as "the effect of rainfall variability".

(8) 59 – grammar problem

Response#8:

We rephrased the sentence into "*anthropogenic amplification of rainfall variability has been identified [Zhang et al., 2024]*".

(9) 72 –"a major"

Response#9:

We corrected accordingly.

(10) 59-78 – the paragraph starts with climate variability and ends with nitrate. I recommend keeping to one topic per paragraph.

Response#10:

Thanks for the suggestion. We rephrased the content accordingly into three paragraphs.

(11) 99-100 sentence fragment

Response#11:

Thanks! The sentence was modified into "*Notably, the 2018 event triggered unprecedented tree mortality across multiple species in Central European forests, accompanied by unexpectedly persistent drought legacy effects [Schuldt et al., 2020]*" (Lines 100-102)."

Reference:

*Schuldt, B., Buras, A., Arend, M., Vitasse, Y., Beierkuhnlein, C., Damm, A., Gharun, M., Grams, T. E. E., Hauck, M., Hajek, P., Hartmann, H., Hiltbrunner, E., Hoch, G., Holloway-Phillips, M., Körner, C., Larysch, E., Lübbe, T., Nelson, D. B., Rammig, A., Rigling, A., Rose, L., Ruehr, N. K., Schumann, K., Weiser, F., Werner, C., Wohlgemuth, T, Zang, C. S., & Kahmen, A.: A first assessment of the impact of the extreme 2018 summer drought on Central European forests, Basic Appl. Ecol., 45, 86–103, https://doi.org/10.1016/j.baae.2020.04.003, 2020.*

(12) 105 – "it is"

We corrected accordingly.

(13) 107 – The objectives are somewhat broad. It has been established that precipitation variability can affect N dynamics, and it would help to be more specific in this paragraph about the aspects of variability being tested and what if anything has been done to address them previously. In other words, how the specific objectives of this study relate to gaps in knowledge left by previous studies?

Response#13:

Thanks for the suggestion. The paragraph was rephrased as "*In order to fill the gap, the present study explored the impact of rainfall variability on N dynamics and its potentially negative influence on water quality across inter-annual and intra-annual timescales. To characterize rainfall variability, a stochastic rainfall generator [Robinson & Sivapalan, 1997] was employed to generate rainfall time series with different climatic characteristics. The research was conducted in a small agricultural catchment located in Central Germany, where a hydrological model was previously established utilizing the fully coupled surface-subsurface numerical simulator HydroGeoSphere [Yang et al., 2018]. The framework of N dynamics was modified from the ELEMeNT approach [Exploration of Long-tErM Nutrient Trajectories; Van Meter et al., 2017]. The research is divided into two main components. First, three representative years (with high, normal, and low annual precipitation amounts, respectively) were chosen from the past two decades in Central Germany as the target scenarios. A fourth scenario with low annual precipitation amounts coupled with reduced plant uptake represents a case where vegetation is partially destroyed by extreme droughts. The statistical analyses of nitrogen loads and fluxes and a comparison across different scenarios were conducted to reveal the influence of inter-annual rainfall variability. Second, rainfall time series generated using the stochastic rainfall generator by separately altering specific parameters were used to substitute the rainfall data in the simulation period to drive the flow and nitrogen transport*

*models. The responses of the N loads and fluxes to the parameters (e.g., the amplitudes of the seasonal variations in the storm durations and inter-storm period) were thoroughly analyzed to clarify the effect of intra-annual rainfall variability. The study will provide theoretical support to formulate fertilization strategies and protect aquatic ecosystems under climate change in the future.”*

(14) 163: cross section is not discernable and it is not clear what is the source of the saturation values

Response#14:

Thanks for pointing that out. We modified Figure 1 and its caption accordingly:

[Figure]

**Figure 1.** (a) The catchment 'Schäfertal' indicated by the red line (background image from © Google Maps), with a cross-sectional view for the flow and saturation [*Yang et*

*al.*, 2018]. (b) The distribution of soil type in the catchment. (c) The measured daily precipitation (P), discharge (Q), and the simulated actual evapotranspiration (ET) [*Yang et al.*, 2018].

(15) 164: this figure is mostly recycled from Wang et al., 2023 but no citation is given.

Response#15:

We added the reference "Yang et al. [2018]" accordingly.

(16) 169: Please clarify if/how these data are used in the current study.

Response#16:

Thanks for the suggestion.

We clarified that by adding " The $C_Q$ and N surplus are used to calibrate the N transport model [Wang et al., 2023]".

(17) 192 – delete "in details"

Response#17:

We corrected accordingly.

(18) 231 – what does this mean that the calibrated model was "verified" over the entire simulation period?

Response#18:

Thanks! We rephrased the sentence into "*Subsequently, the calibrated model was verified by reproducing time-variable groundwater levels for the wells over the entire simulation period [Yang et al., 2018].*"

(19) 240 – meaning unclear "delineated corresponding to the reality"

Response#19:

Thanks! We have removed the unnecessary sentences.

(20) 245-6: "in route" grammar

Response#20:

Thanks for pointing that out. The sentence was modified to "Throughout the nitrogen cycle, various forms of nitrogen undergo complex biogeochemical processes".

(21) 258-260: meaning unclear.

Response#21:

We rephrased the sentence into "The framework is able to capture the main processes of nitrogen transformation and transport in soil and groundwater [Yang et al., 2018]"

(22) 320: "validation" might not be the right word (not the same as calibration).

Response#22:

Thanks for pointing that out. The sentence was modified to "Subsequently, the calibrated model was verified by reproducing time-variable groundwater levels for the wells over the entire simulation period [Yang et al., 2018]".

(23) 323: Wang et al 2023 refers readers to Yang 2018 for more details and is not an easy source of information about the estimation of the N cycling parameters, uncertainty of those parameters or the quality of fit to the data. These are crucial aspects of the calibrated model and should be presented clearly and succinctly for the readers.

Response#23:

Thanks for the suggestion. The brief calibration results were added in the beginning of result section as "*The calibrated N transport model showed good performance in fitting the in-stream nitrate concentration (Figure 3a), with a Nash-Sutcliffe efficiency (NSE) value of 0.70. Subsequently, the calibrated model was validated over the entire simulation period from 1997 to 2010 (~1 day of CPU time), with a NSE value of 0.55. Due to incomplete observation and the high CPU time, the NSE is still considered to be acceptable. The calibrated best-fit values for the transport parameters are listed in Table S1 [see in Supporting Information]. More detailed calibration results refer to Wang et al. [2023]*".

(24) 325: "impermeable for nitrate" (and water?)

Response#24:

We rephrased the sentence as "The bedrock is treated as impermeable for water and nitrate".

(25) 333-375: probably don't need this much detail about the rainfall generator

Response#25:

Thanks for the suggestion. The section 3.3 has been simplified.

(26) 336: "a stochastic model"

Response#26:

We corrected accordingly.

(27) Table 1 – This table has too many numbers and variables for readers to easily absorb. Consider placing with a schematic, examples, or another simpler figure or table.

Response#27:

Thanks for the suggestion. We updated table 1 accordingly.

(28) 399-401 – grammar problem, meaning is lost

Response#28:

Thanks for pointing that out. The sentence has been modified to "In order to explore the effect of intra-annual rainfall variability on N dynamics, the linear regression analyses between the parameters of the stochastic rainfall generator and N loads, fluxes, as well as $C_Q$ were conducted, in which the parameters of NY serve as a reference".

(29) 458 – Why is soil denitrification lumped with GW denitrification? Are they expected to be similar?

Response#29:

Thanks for point that out. Soil denitrification and GW denitrification are identical in mechanism, and we expected to see how much nitrogen in total is consumed by denitrification under different rainfall pattern. Therefore, they are added together in the study.

(30) 503 – These figures are confusing because the response variables (SON, SIN, LEA, Cq) are not on the z-axes.

Response#30:

Thanks for the suggestion! Figure 8 and Figure 9 are presented in the same way. We will use Figure 8 as an example. In the stochastic rainfall generator, the distributions of the average storm duration and the average inter-storm period over the course of a year are depicted by Sine functions (S3 & S4 in the Supporting Information), whose characteristics are determined by the amplitudes of the seasonal variations in the average storm duration and the average inter-storm period ($\alpha_\gamma$ and $\alpha_\delta$) reflecting climate change, when other parameters keep invariable. As for a certain $\alpha_\gamma$, there is a specific distribution of the average storm duration over the course of a year, which is

presented in the x-z plane. The color of the panels represents the value of $\alpha_\gamma$. With $\alpha_\gamma$ decreasing, the distribution of the average storm duration transitions and the year shifts from wet to dry. When the year becomes drier, the annual SON load increases, while the SIN load, leaching flux, and in-stream nitrate concentration decreases on the y-axes. Therefore, the transformation and transport of N are subject to retardation in a dry year when storms with shorter duration occur mid-year.

(31) 564: 5.1 section title: consider being more specific about what increased rainfall does to the N dynamics

Response#31:

Thanks for the suggestion. We changed section tile of 5.1 into " N transformation and transport upgrade in wet years ".

(32) 608: It seems notable that the high flows during the 2018 to 2019 drought are as high as the high flows from 2014-2018, and the main difference during 2018-2019 seems to be in the low flow periods.

Response#32:

Thanks for pointing that out. Yes, as you observed, droughts mainly occurred in the middle of the year, resulting in extreme low flow (Figure 12). During the low-flow periods, nitrogen absorbed by vegetation in the growth stage is significantly reduced and nitrogen (SON & SIN) starts to accumulate in the soil. The accumulated nitrogen was subsequently flushed out during the rewetting period [Winter et al., 2023].

Reference:

Winter, C., Nguyen, T., Musolff, A., Lutz, S., Rode, M., Kumar, R., Fleckenstein, J.: Droughts can reduce the nitrogen retention capacity of catchments, Hydrol. Earth Syst. Sci., 27(1):303-318, https://doi.org/10.5194/hess-27-303-2023, 2023.

(33) 670: 5.6. Consider discussing: Data limitations, uncertainty of parameters, model process representations

Response#33:

Thanks for the suggestions.

We added the discussion to address the limitation in data, model representation and

parameter uncertainty as "*In addition, the spatial variation of denitrification rate coefficient may lead to uncertainty in the simulated results. Thus, the effect of temperature on denitrification rate should be investigated in detail and depicted in the framework of N dynamics. Due to the lack of groundwater nitrate concentration, the nitrate transport model was calibrated using only in-stream nitrate concentration data [Wang et al., 2023]. More nitrate concentration data in groundwater and river will contribute to establish a better nitrate transport model.*"

(34) Table S1 – van Meter reference is missing date; bibliography is not included in this document

Response#34:

Thanks for pointing that out. We have corrected the table S1:

**Table S1**. The parameters for the N source zone. The parameters were adjustable and calibrated (referring to Yang et al. [2022]).

| Parameter | Process | Adjustable range | Reference | Value of best fit |
|---|---|---|---|---|
| $k_a$ | Mineralization | 0~0.7 | Van Meter et al. [2017] | 0.0109 day$^{-1}$ |
| $k_p$ | Mineralization | 0~0.7 | Van Meter et al. [2017] | 0.0008 day$^{-1}$ |
| $p1$ | Plant uptake | 60~160 | Van Meter et al. [2017] | 160 kg·ha$^{-1}$ |
| $p2$ | Plant uptake | 1~10 | | 10 kg·ha$^{-1}$ |
| $p3$ | Plant uptake | 1~60 | | 24 day |
| $\lambda_s$ | Denitrification in soil | 0~0.7 | Nguyen et al. [2021] | 0.0003 day$^{-1}$ |
| $k_l$ | Leaching | 1~1000 | | 3.3906 day$^{-1}$ |
| $q_{ref}$ | Leaching | 1e$^{-4}$~1 | | 0.0654 m·day$^{-1}$ |
| $\lambda$ | Denitrification in water | 1e$^5$~1e$^{-1}$ | Nguyen et al. [2021] | 0.0088 day$^{-1}$ |

---

## Author Comment (AC4)

Dear reviewer,

Thanks you so much for your constructive comments in the discussion phase. Please check our responses to all your comments point by point (followed and marked in blue). However, the revised manuscript will be provided in the next step of the review processes.

Best regards

**Responses:**

The authors present an original piece of research focusing on the catchment response of N fluxes to rainfall interannuel and intra-annual variability using synthetic experiments based on the Hydrogeosphere model. I found the study clean and rigorously described, calibration method is sensible. Figures provide useful and clear illustrations of the results. The manuscript is well written. I think the discussion could be expanded and therefore I recommend a minor revision of the manuscript.

Especially I would have **2 comments on the discussion**:

1) several parts of the discussion present some reactions as simulation results when they are a direct consequence of the modelling equations. Ex.: Lines 566 to 568, the impact of temperature and wetness on mineralization is constructed mathematically in equations 1 and 2, isn't it? Same comment regarding plant uptake and denitrification. According to my opinion, the interest of the model is rather to calculate which of these mechanisms is going to dominate the response, and also it helps to consider different time scales of response, which is particularly relevant for droughts (cf. lines 97-99)

Response #1:

Thank you for the comment. We truly agree that the reaction is a direct consequence of the mechanisms that are built in the equations already. That mean, some trends / patterns can be simply known even without actually performing the simulations. This study is aimed to investigate how the rainfall variability influences the N transport and which

process is the key control of that influences. E.g, we found that the plant-uptake is the main factor caused the high in-stream concentration in extreme dry years.

In response to your comment, we have revised the discussion to clarify the dominant mechanisms governing in-stream nitrate concentration responses as "*The comparison of $C_Q$, N loads, and N fluxes among the four scenarios of the inter-annual rainfall variability reveals its effect on N dynamics and water quality. Mineralization, the crucial process of the transformation from SON to SIN, is highly sensitive to soil wetness. Consequently, the highest average mineralization rates, observed in WY scenario, promote the transformation of N, including plant uptake, denitrification, and leaching. In contrast, low mineralization in DY scenario leads to SON accumulation and thereby restraining overall N dynamics*".

2) I have a general comment on the modelling choice that is N mechanisms are much more simplified than the water processes. While I am very aware of the computationnal challenges associated with such virtual experiments, I find intellectually disturbing to have a fully mechanistic approach to represent water combined with a representation of nitrogen very simplified in comparison. What do you think?

Response #2.1:

Thank you for this comment! We totally agree with your point that the N processes represented in the model is very simple compared with the groundwater flow. The complexities N fluxes in source zone were simplified by defining a framework describing the main pathway for N-NO$_3$ leachate with temporally constant external N input. On the one hand, this simplification neglected other processes such as the time-variant external N input, direct input of external N to inorganic N pool, the transformation from protected organic N to active organic N, and the loss of organic N via dissolution. On the other hand, this simplification allows us to focus on the main source of N-NO$_3$, rather than keeping track of the full nitrogen fluxes in the source zone of the catchment while maintaining the overall nitrogen balance using surplus as a constraint. In this sense, we think it is an effective tool to answer certain questions, for example in our study, which process controls the high nitrate concentrations in river

while dry climate.

To response to this point, we discuss the limitation of the simplified N framework additionally in section 5.6 in the revised manuscript.

[Figure]

**Figure 2.** The framework simulating the nitrogen transport in soil and groundwater, modified based on [Yang et al., 2021].

There is no representation of plant-soil processes, and considering the results of the EDY scenario in terms of N dynamics, the response of plants growth or death to water stress seems to be a key mechanism.

Response #2.2:

Thanks for pointing that out! Yes, the plant-soil process is not represented in the model. This means, in the extreme dry scenario, the plants death was not actually simulated, but manually being assigned a lower N-uptake potential. The plant growth stages were assumed to be constant (for example, the same N uptake potential in different). This is also due to lack of representation for the plant-soil processes in the simplified N framework. N uptake process is only mathematically described using the empirical formula (Equation 6).

However, such simplification was proved to be effective in term of reproducing the

reasonable N loads and in-stream nitrate concentrations. It can still be used to identified the key effect of plant-die-off (i.e. reduced N uptake potential) on stream water quality during extreme dry climate, comparing with other process such as mineralization and leaching.

To response to this point, we discuss the limitation of neglecting the plant-soil processes in section 5.6 in the revised manuscript.

Also, about the fact that all external nitrogen inputs are introduced in the SON pool (lines 250, 252), I was wondering to what extend it refers to a reality? Are fertilizers mainly applied as urea?

Response #2.3:

Thanks for pointing that out!

The external N input represents atmospheric deposition, biological fixation, animal manure from the pasture area, and N fertilizer from the farmland. It does include organic and inorganic inputs. In our framework based on ELEMeNT model [Van Meter et al., 2017], It is assumed that all external nitrogen inputs are introduced in the organic pool. This assumption is made based on the fact that most of the nitrate ($N-NO_3$) fluxes from source zone has undergone biogeochemical transformation in organic N pool (Haag and Kaupenjohann, 2001). The described framework simplifies complexities of different N pools and transformations via mineralization, dissolution, and denitrification within the soil zone, while preserving the main pathway for $N-NO_3$ leachate. In this sense, assumption that the external N input contributes only to the SON is acceptable in the study.

To response to this point, we discuss the limitation of the simplified N framework additionally in section 5.6 in the revised manuscript (also see our response above).

Also, I did not understand why is this framework acceptable and according to which criteria or arguments (line 258-260)?

Response #2.4:

Thanks for the comment!

Although the mechanisms of N dynamics in this framework are quite simplified, we

think it is acceptable for our study because:

1) It is effective in term of reproducing the reasonable N loads and in-stream nitrate concentrations in the catchment. The modeled in-stream concentrations fit well with the observations with a Nash-Sutcliffe efficiency (NSE) value up to 0.70 [Wang et al., 2023]. The simulated N mass balance was also meaningful: with an external N input of 180 kg ha/yr, the total load of SON and SIN were 468 kg ha$^{-1}$ and 34 kg ha$^{-1}$, respectively. The former accounted for 93% of the total N in the soil, being consistent with the values reported in Stevenson (1995) (organic N fraction > 90%). The mineralization rate was 179 kg ha$^{-1}$ yr$^{-1}$, which is within the range reported by Heumann et al. (2011) for study sites in Lower Saxony, Germany ([14 - 187] kg N / ha yr$^{-1}$). 67% of the SIN was absorbed by plants at a rate of 120 kg ha$^{-1}$ yr$^{-1}$, which is comparable to the value suggested in Nguyen et al. (2021) for the same area (around 120 kg ha$^{-1}$ yr$^{-1}$). The denitrification flux was 45 kg ha$^{-1}$ yr$^{-1}$ (4 kg ha$^{-1}$yr$^{-1}$ in the soil, 41 kg ha$^{-1}$ yr$^{-1}$ in the groundwater), which is within the range (8 - 51) kg ha$^{-1}$ yr$^{-1}$ reported in Hofstra and Bouwman (2005) for 336 agricultural areas around the world. The SIN leached into the groundwater at a rate of 55 kg ha$^{-1}$ yr$^{-1}$,which is comparable to the range (15 – 60kg N ha$^{-1}$ yr$^{-1}$) reported in Nguyen et al. (2021). Finally, approximately 25% of the leachate was exported to the stream with the remaining 75% being removed by denitrification during the transport in the groundwater, which are comparable to the values 39% and 61% reported in Nguyen et al. (2021).

2) The described framework simplifies complexities of different N pools and transformations via mineralization, dissolution, and denitrification within the soil zone, while preserving the main pathway for N-NO$_3$ leachate. In this sense, the framework can still be used to explore the influence of rainfall on water quality vie the key processes (mineralization, plant-uptake and leaching).

To response this point, we briefly describe the simulated N loads and in-stream nitrate concentrations, by comparing with observations or other literatures (which has been done in our previous Wang et al., 2023) in the revised manuscript.

**I have also a few minor comments below:**

3) line 41: I suspect that fertilization strategies will evolve with the changes in timing of plant growth stages with warming

Response #3:

Thanks! This is a very good point. We agree that the fertilization strategies may evolve with warming. However, in this study, the plant growth stages were assumed to be the same for different years (for example, the same sowing date in either wet year or dry year). This is due to lack of representation for the plant-soil processes in the model. Such assumption was proved to be effective in term of reproducing the reasonable N loads and in-stream nitrate concentrations [Yang et al., 2021], however, may leads to uncertainties in the simulated N fluxes.

4) introduction should maybe be restricted to context elements that are directly linked to the study, lines 45 to 61: it could go more directly to the point of rainfall variability skipping the details on extremes. Lines 79 to 82: I missed the link with the present study here.

Response #4:

Thank you for your suggestions. We have revised the first paragraph of the Introduction as "*The hydrological processes are susceptible to meteorological conditions on various spatial and temporal scales [Ionita et al., 2017; Laaha et al., 2017; Zhang et al., 2021]. In the past decades, extreme climate events intensified by human-induced climate change have frequently occurred globally [Pall et al., 2011; Min et al., 2011; Williams et al., 2015; Hari et al., 2020], most of which caused water scarcity and poor water quality at regional scales [Zwolsman and van Bokhoven, 2007; Delpla et al., 2009; Whitehead et al., 2009; Stahl et al., 2016; Ballard et al., 2019; Bauwe et al., 2020; Geris et al., 2022]. Heavy rainstorms and severe droughts being the predominant extreme climate events around the globe share the common characteristic of rainfall variability [Trenberth et al., 2011; Pendergrass et al., 2017; Hanel et al., 2018]. In the context of global warming scenarios, rainfall variability has been amplified*

*anthropogenic [Zhang et al., 2024]. Thus, the effect of rainfall variability on water resources has attracted much attention around the world" (Line 47-59)".*

We also deleted the following sentences describing historical extreme events, which are not directly relevant to this study *"Extreme rainfall events in 2002, 2013, and 2021 caused significant threats to human safety and substantial damage to the environment, economy, and infrastructure [Ulbrich et al., 2003; Thieken et al., 2016; Voit & Heistermann, 2024]. Three notable summer droughts occurred in 2003, 2015, and 2018–2019 (consecutive), driven by precipitation deficits combined with temperature anomalies during the growing season [Fink et al., 2004; Schär et al., 2004; Ciais et al., 2005; Orth et al., 2016; Hanel et al., 2018; Hari et al., 2020; Camenisch et al., 2020].*"

5) lines 145-146: at which frequency are concentration measured?

Response #5:

Thanks! We clarified this by adding "..at 14-days to monthly intervals [Dupas et al., 2017], covering the period 2001–2010."

6) line 339-340 what is a "good performance under Climate change" for the stochastic generator?

Response #6:

Thank you for raising this point.

Wilusz et al. [2017] coupled the rainfall generator [Robinson & Sivapalan, 1997], a rainfall-runoff model [Kirchner, 2009], and the rank StorAge Selection (rSAS) transit time model [Harman, 2015] to decompose the relationship between rainfall variability and the time-varying fraction of young water (<90 days old) in streams (FYW). The rainfall generator was specifically employed to synthesize daily rainfall time series with statistical properties that are identical with or systematically different from historic rainfall. Crucially, the generator is capable of representing individual storm events, between-storm variability, within-storm variability, and seasonality [Robinson &

Sivapalan, 1997]. Thus, based on these capabilities, the sentence was modified into "*It outputs rainfall series representing different rainfall patterns under climate change*".

7) Figure 4 a: it would be useful to have measured C_Q in the plot too. The legend refers to "acceptable simulations" but so far as I understand it is more the variability associated to the generator in each scenarios (n=100) isn't it?

Response #7:

Thank you for the suggestion!

We have added the measured in-stream nitrate concentration data to Figure 3a. Indeed, gray areas in Figure 4(b1)~(b4) are formed by the 100 realizations. We changed the caption of Figure 4 to clarify this as "*The grey areas are formed by the realizations of each scenario*".

8) lines 577-578: are preferential flowpaths represented in the model?

Response #8:

Thank you! Preferential flow paths on top of the aquifer (fast shallow flow paths) has been identified and discussed in our previous study [Yang et al., 2018], plotted in Figure 1a. Such fast shallow flow paths established a hydraulic connection between channel and the slope top in the wet seasons. We added "(I-I cross section, Figure 1a)" to refer to the flow paths.

[Figure]

**Figure 1.** (a) The catchment 'Schäfertal' indicated by the red line (background image from © Google Maps), with a cross-sectional view for the flow and saturation [*Yang et al*., 2018]. (b) The distribution of soil type in the catchment. (c) The measured daily precipitation (P), discharge (Q), and the simulated actual evapotranspiration (ET) [*Yang et al*., 2018].

9) it would be useful to see the effect on water fluxes as well: especially recharge flux for groundwater and Actual ET in Figure 10 or Figure S2 (lines 653 to 656)

Response #9:

Thank you for your suggestion. We added the rechange fluxes (Rainfall-actual ET) and the actual ET in Figure S2 of the supporting information.

---

## Author Response (AR1)

Dear editor and reviewers,

Thank you so much for your constructive comments in the discussion phase. Please check our responses to all your comments point by point (followed and marked in blue). Unless specified, all the line numbers in our responses refer to the line numbers of the TRACK CHANGES version of the manuscript.

Beside the responses to the two reviewers, we also made the following updates to the manuscript:

- We made grammatical edits throughout the manuscript.
- Due to a minor mistake in the simulations of our last version manuscript, we re-performed all the numerical simulations. The following changes / unchanged contents should be noted:
  - 1. Necessary updates were made to the text describing the new results.
  - 2. New results do not alter the main conclusions of this study. The new simulation results showed changes in the absolute values of N fluxes and loads. However, the variation patterns across scenarios (e.g., differences among WY, NY, DY, EDY) persisted. The effects of varying monthly average storm duration distributions and inter-storm period distributions on SON, SIN loads, leaching flux, and in-stream nitrate concentrations were also consistent. Therefore, the main conclusions of this study remain unchanged.
  - 3. New figures (Figure 3 Figure 13) were re-regenerated based on the new simulation.
  - Parameter Pdrizzle (probability of drizzle events) was additionally discussed as its effect on N dynamics were newly identified (lines 724-742, 883-894).
- We revised the document format according to the requirements.
- We added an authors' affiliation (Lines 8-9).

**Responses:**

**RC1:**

This manuscript presents a numerical modeling study exploring how inter- and

intra-annual precipitation variability affects nitrogen (N) loads and fluxes in a catchment. The topic is timely and important for water quality and environmental management. The paper is generally well written, though some grammatical edits are needed. The methods and results are mostly clear and logically presented. The conclusions are relevant and likely to be of interest to scientists and resource managers focused on mitigating nutrient pollution.

**Primary suggestions are to:**

(1) Clarify and narrow the research focus: While the broad importance of precipitation variability is well established in the introduction, the final introductory paragraph should better define the unique contribution this work makes to the literature. Consider making the statement of objectives more specific and clarifying how the objectives fill gaps left by other recent studies.

**Response #1:**

Thanks for point that out! We have rephrased the last paragraph of the introduction part to define the unique contribution this work as "To fill the gap, the present study explored the impact of rainfall variability on N dynamics and its potential influence on water quality across inter-annual and intra-annual timescales. To characterize rainfall variability, a stochastic rainfall generator [Robinson and Sivapalan, 1997] was employed to generate rainfall time series with different climatic characteristics. The research was conducted in a small agricultural catchment located in Central Germany, where a hydrological model was previously established utilizing the fully coupled surface-subsurface numerical simulator HydroGeoSphere [Yang et al., 2018]. The framework of N dynamics was modified from the ELEMeNT approach [Exploration of Long-tErM Nutrient Trajectories; Van Meter et al., 2017]. The research is divided into two main components. First, three representative years (with high, normal, and low annual precipitation amounts, respectively) were chosen from the past two decades in Central Germany as the target scenarios. A fourth scenario with low annual precipitation amounts coupled with reduced plant uptake represents a case where vegetation is partially destroyed by extreme droughts. The statistical analyses of N loads and fluxes and a comparison across different scenarios were

conducted to reveal the influence of inter-annual rainfall variability. Second, rainfall time series generated using the stochastic rainfall generator by separately altering specific parameters were used to substitute the rainfall data in the simulation period to drive the flow and nitrogen transport models. The responses of the N loads and fluxes to the parameters (e.g., the amplitudes of the seasonal variations in the storm duration and inter-storm period) were thoroughly analyzed to clarify the effect of intra-annual rainfall variability. The study will provide theoretical support for formulating fertilization strategies and protecting aquatic ecosystems in the context of climate change" (Lines 133-163).

**Reference:**

Robinson, J., & Sivapalan, M.: Temporal scales and hydrological regimes: Implications for flood frequency scaling, Water Resour. Res., 33(12): 2981-2999, https://doi.org/10.1029/97WR01964, 1997.

Yang J., Heidbüchel, I., Musolff, A., Reinstorf, F., and & Fleckenstein. J. H.: Exploring the dynamics of transit times and subsurface mixing in a small agricultural catchment, Water Resour. Res., 54, https://doi.org/10.1002/2017WR021896, 2018.

Van Meter, K. J., Basu, N. B., Van Cappellen, P.: Two centuries of nitrogen dynamics: Legacy sources and sinks in the Mississippi and susquehanna river basins, Global Biogeochem. Cycles, 31 (1), 2–23., https://doi.org/10.1002/2016GB005498, 2017.

(2) Differentiate from previous work: The study references and builds on previous work, particularly Wang et al. (2023). However, the manuscript does not always make it clear where prior work ends and the current study begins. The manuscript also refers the reader to the previous studies for some crucial details (e.g. calibration to stream nitrate concentrations), which are not easy to find in the previous studies. I recommend adding further descriptions of the related studies at this site and delineating which analyses, model developments, and findings are new in this study.

**Response#2:**

**Thanks for the points!**

Generally, this study employed the model developed in our previous work Wang et al.

[2023] for a same study site. The model setup, method for simulating the flow and N transport and calibration, part of parameters are all from Wang et al. [2023]. However, new contributions are attributed to exploring the effect of rainfall variability on N dynamics. To clarify the existing work and the new contributions, we rephased the method section such that the new contributions are highlighted as follow: "Based on the hydrological and transport model, the effect of topographic slope on the export of nitrate [Yang et al., 2022] and the spatial-temporal variation of nitrogen retention [Wang et al., 2023] were explored. In the present study, the two models were employed to investigate the effect of rainfall variability on N dynamics, for which rainfall time series with climatic characteristics, substituting for the precipitation data during the simulation period, were generated by a stochastic rainfall generator [Robinson and Sivapalan, 1997; Wilusz et al., 2017]" (Lines 235-241).

Additionally, more details regarding the model calibration were added in the results section as a short review: "The calibrated N transport model showed good performance in fitting the in-stream nitrate concentration (Figure 4a), with a Nash-Sutcliffe efficiency (NSE) of 0.79. The modeled N surplus of 51.87 kg ha-1 yr-1 is comparable to the measured value of 48.8 kg ha-1 yr-1. The calibrated best-fit values for the transport parameters are listed in Table S1 (see Supporting Information)" (Lines 487-491).

(3) Clarify model-data connection: A clearer connection could be made between the numerical model and real-world observations. The use of simple abstractions in the nutrient transport component of the model is reasonable, but also requires careful consideration of (a) how well the simplified processes representations mimic actual processes, (b) the uncertainty of the parameter estimates, and (c) the accuracy of the model in terms of reproducing observations. Otherwise, there is a risk of circularity: the model is built around certain processes and parameterizations, and then used to test the importance of those same processes and parameters.

**Response#3:**

Thanks for the suggestions. We made the following changes to clarify model-data connection:

1. We added the observed in-stream nitrate concentrations along with the simulated ones (see the figure below) such that they can be clearly compared.

**Figure 4.** The fluctuations of simulated in-stream nitrate concentration ( $C_Q$ ) under (a) the historical rainfall (1997-2007) and the scenarios of ( $b_1$ ) WY, ( $b_2$ ) NY, ( $b_3$ ) DY, and ( $b_4$ ) EDY (2008-2010). The grey areas are formed by the realizations of each scenario.

We added a description of the calibration results to show the accuracy of the model in terms of reproducing observations as "The calibrated N transport model showed good performance in fitting the in-stream nitrate concentration (Figure 4a), with a Nash-Sutcliffe efficiency (NSE) of 0.79. The modeled N surplus of 51.87 kg ha-1 yr-1 is comparable to the measured value of 48.8 kg ha-1 yr-1. The calibrated best-fit values for the transport parameters are listed in Table S1 (see Supporting Information). Figure 3 illustrates the 14-year N mass balance simulated by the calibrated N transport model in the entire catchment. In the soil source zone, the total N consisted of SON (552 kg ha-1, including SON(a) of 90 kg ha-1 and SON(p) of 462 kg ha-1) and SIN (48 kg ha-1). The load of SON accounts for 92% of the total N, which corresponds to the research result that the organic N fraction is greater than 90% [Stevenson., 1995]. As for N transformation, the mineralization rate of 173 kg ha-1 yr-1 is within the range (14-187 kg ha-1 yr-1) reported by Heumann et al. [2011] for study sites located in Germany. 71% of the SIN was absorbed by plants at a rate of 123 kg ha-1

yr-1, which is very close to the value (around 120 kg ha-1 yr-1) suggested in Nguyen et al. [2021] for the same area. The denitrification flux of 38 kg ha-1 yr-1 (4 kg ha-1 yr-1 in the soil source zone, 34 kg ha-1 yr-1 in the groundwater) is within the range (8-51 kg ha-1 yr-1) investigated for 336 agricultural areas around the world by Hofstra and Bouwman [2005]. The SIN entered the groundwater zone at a rate of 44 kg ha-1 yr-1, which is within the range (15-60 kg N ha-1 yr-1) reported in Nguyen et al. [2021]. Approximately 23% of the leachate of SIN was exported to the stream, where the rest of the leachate (77%) was removed by denitrification. The export and denitrification fluxes are comparable to the values 39% and 61% reported in Nguyen et al. [2021]. Based on measured groundwater chemistry data from nearby catchments and multiple estimates of denitrification potential that have been proved and reported [Hannappel et al., 2018], we conclude that the simulated transformation and transport of N are acceptable.

*Figure 3.* The simulated N loads and fluxes in the entire catchment." (Lines 487-514).

2. We added short discussion about the uncertainty of the parameter in the section of Discussion as "As a complex biogeochemical process, denitrification is governed by various factors such as temperature, soil moisture content, and SIN load, rather than a first-order decay process. Nonetheless, the current transport model does not account for spatial heterogeneity in denitrification. These simplifications may

Line-by-Line comments:

(4) Line 14: "performance" is wrong word. "effect"?

Response#4:

Thanks! We changed "performance" into "manifestation" (Line 17).

(5) 34: SON not defined at this point

Response#5:

Thanks, we rephrased as ".. soil organic nitrogen (SON)..." (Lines 41-42).

(6) 39 – "a small effect"

Response#6:

We corrected accordingly (Line 47).

(7) 58 – not clear what "their" refers to

Response#7:

We clarified it as "the effect of rainfall variability" (Line 79).

(8) 59 – grammar problem

Response#8:

We rephrased the sentence into "anthropogenic amplification of rainfall variability has been identified [Zhang et al., 2024]" (Lines 75-79).

(9) 72 - "a major"

Response#9:

We corrected accordingly (Line 95).

(10) 59-78 – the paragraph starts with climate variability and ends with nitrate. I recommend keeping to one topic per paragraph.

Response#10:

Thanks for the suggestion. We rephrased the content accordingly into three paragraphs (Lines 71-101).

(11) 99-100 sentence fragment

Response#11:

Thanks! The sentence was modified into "Notably, the 2018 event triggered

unprecedented tree mortality across multiple species in Central European forests, accompanied by unexpectedly persistent drought legacy effects [Schuldt et al., 2020]" (Lines 112-114).

**Reference:**

Schuldt, B., Buras, A., Arend, M., Vitasse, Y., Beierkuhnlein, C., Damm, A., Gharun, M., Grams, T. E. E., Hauck, M., Hajek, P., Hartmann, H., Hiltbrunner, E., Hoch, G., Holloway-Phillips, M., Körner, C., Larysch, E., Lübbe, T., Nelson, D. B., Rammig, A., Rigling, A., Rose, L., Ruehr, N. K., Schumann, K., Weiser, F., Werner, C., Wohlgemuth, T., Zang, C. S., & Kahmen, A.: A first assessment of the impact of the extreme 2018 summer drought on Central European forests, Basic Appl. Ecol., 45, 86–103, https://doi.org/10.1016/j.baae.2020.04.003, 2020.

(12) 105 - "it is"

**Response#12:**

**We corrected accordingly (Line 131).**

(13) 107 – The objectives are somewhat broad. It has been established that precipitation variability can affect N dynamics, and it would help to be more specific in this paragraph about the aspects of variability being tested and what if anything has been done to address them previously. In other words, how the specific objectives of this study relate to gaps in knowledge left by previous studies?

**Response#13:**

Thanks for the suggestion. The paragraph was rephrased as "To fill the gap, the present study explored the impact of rainfall variability on N dynamics and its potential influence on water quality across inter-annual and intra-annual timescales. To characterize rainfall variability, a stochastic rainfall generator [Robinson and Sivapalan, 1997] was employed to generate rainfall time series with different climatic characteristics. The research was conducted in a small agricultural catchment located in Central Germany, where a hydrological model was previously established utilizing the fully coupled surface-subsurface numerical simulator HydroGeoSphere [Yang et al., 2018]. The framework of N dynamics was modified from the ELEMENT

approach [Exploration of Long-tErM Nutrient Trajectories; Van Meter et al., 2017]. The research is divided into two main components. First, three representative years (with high, normal, and low annual precipitation amounts, respectively) were chosen from the past two decades in Central Germany as the target scenarios. A fourth scenario with low annual precipitation amounts coupled with reduced plant uptake represents a case where vegetation is partially destroyed by extreme droughts. The statistical analyses of N loads and fluxes and a comparison across different scenarios were conducted to reveal the influence of inter-annual rainfall variability. Second, rainfall time series generated using the stochastic rainfall generator by separately altering specific parameters were used to substitute the rainfall data in the simulation period to drive the flow and nitrogen transport models. The responses of the N loads and fluxes to the parameters (e.g., the amplitudes of the seasonal variations in the storm duration and inter-storm period) were thoroughly analyzed to clarify the effect of intra-annual rainfall variability. The study will provide theoretical support for formulating fertilization strategies and protecting aquatic ecosystems in the context of climate change" (Lines 133-163).

(14) 163: cross section is not discernable and it is not clear what is the source of the saturation values

Response#14:

Thanks for pointing that out. We modified Figure 1 and its caption accordingly [Lines 205-212]:

Figure 1. (a) The catchment 'Schäfertal' indicated by the red line (background image from © Google Maps), with a cross-sectional view for the flow and saturation [Yang et al., 2018]. (b) The distribution of soil type in the catchment. (c) The measured daily precipitation (P), discharge (Q), and the simulated actual evapotranspiration (ET) [Yang et al., 2018].

(15) 164: this figure is mostly recycled from Wang et al., 2023 but no citation is given.

**Response#15:**

We added the reference "Yang et al. [2018]" accordingly (Line 210).

(16) 169: Please clarify if/how these data are used in the current study.

**Response#16:**

Thanks for the suggestion.

We clarified that by adding "The  $C_Q$  and N surplus are used to calibrate the N

transport model [Wang et al., 2023]" (Lines 225-226).

(17) 192 – delete "in details"

Response#17:

We corrected accordingly (Line 242).

(18) 231 – what does this mean that the calibrated model was "verified" over the entire simulation period?

Response#18:

Thanks! We rephrased the sentence into "Subsequently, the calibrated model was verified by reproducing time-variable groundwater levels for the wells over the entire simulation period [Yang et al., 2018]" (Lines 283-284).

(19) 240 – meaning unclear "delineated corresponding to the reality"

Response#19:

Thanks! We have removed the unnecessary sentences (Lines 290-296).

(20) 245-6: "in route" grammar

Response#20:

Thanks for pointing that out. The sentence was modified to "Throughout the nitrogen cycle, various forms of nitrogen undergo complex biogeochemical processes" (Lines 299-300).

(21) 258-260: meaning unclear.

Response#21:

We rephrased the sentence into "The framework is able to capture the main processes of nitrogen transformation and transport in soil and groundwater [Yang et al., 2018]" (Lines 312-316).

(22) 320: "validation" might not be the right word (not the same as calibration).

Response#22:

Thanks for pointing that out. This related sentence was deleted. The brief calibration results were added in the beginning of result section as "The calibrated N transport model showed good performance in fitting the in-stream nitrate concentration (Figure 4a), with a Nash-Sutcliffe efficiency (NSE) of 0.79. The modeled N surplus of 51.87 kg

ha-1 yr-1 is comparable to the measured value of 48.8 kg ha-1 yr-1. The calibrated best-fit values for the transport parameters are listed in Table S1 (see Supporting Information)" (Lines 487-491).

(23) 323: Wang et al 2023 refers readers to Yang 2018 for more details and is not an easy source of information about the estimation of the N cycling parameters, uncertainty of those parameters or the quality of fit to the data. These are crucial aspects of the calibrated model and should be presented clearly and succinctly for the readers.

**Response#23:**

Thanks for the suggestion. The calibration result was added in the beginning of result section as "Figure 3 illustrates the 14-year N mass balance simulated by the calibrated N transport model in the entire catchment. In the soil source zone, the total N consisted of SON (552 kg ha-1, including SON(a) of 90 kg ha-1 and SON(p) of 462 kg  $ha^{-1}$ ) and SIN (48 kg  $ha^{-1}$ ). The load of SON accounts for 92% of the total N, which corresponds to the research result that the organic N fraction is greater than 90% [Stevenson., 1995]. As for N transformation, the mineralization rate of 173 kg ha-1  $yr^{-1}$  is within the range (14-187 kg ha-1  $yr^{-1}$ ) reported by Heumann et al. [2011] for study sites located in Germany. 71% of the SIN was absorbed by plants at a rate of 123 kg ha-1 yr-1, which is very close to the value (around 120 kg ha-1 yr-1) suggested in Nguyen et al. [2021] for the same area. The denitrification flux of 38 kg ha-1 yr-1 (4 kg ha-1 yr-1 in the soil source zone, 34 kg ha-1 yr-1 in the groundwater) is within the range (8-51 kg ha-1 yr-1) investigated for 336 agricultural areas around the world by Hofstra and Bouwman [2005]. The SIN entered the groundwater zone at a rate of 44 kg ha-1  $yr^{-1}$ , which is within the range (15-60 kg N ha-1  $yr^{-1}$ ) reported in Nguyen et al. [2021]. Approximately 23% of the leachate of SIN was exported to the stream, where the rest of the leachate (77%) was removed by denitrification. The export and denitrification fluxes are comparable to the values 39% and 61% reported in Nguyen et al. [2021]. Based on measured groundwater chemistry data from nearby catchments and multiple estimates of denitrification potential that have been proved and reported [Hannappel

et al., 2018], we conclude that the simulated transformation and transport of N are acceptable.

*Figure 3.* The simulated N loads and fluxes in the entire catchment." (Lines 492-514).

(24) 325: "impermeable for nitrate" (and water?)

**Response#24:**

We rephrased the sentence as "The bedrock is treated as impermeable for water and nitrate" (Lines 384-385).

(25) 333-375: probably don't need this much detail about the rainfall generator Response#25:

Thanks for the suggestion. The section 3.3 has been simplified (Lines 393-442).

(26) 336: "a stochastic model"

**Response#26:**

We corrected accordingly (Line 395).

(27) Table 1 – This table has too many numbers and variables for readers to easily absorb. Consider placing with a schematic, examples, or another simpler figure or table.

**Response#27:**

Thanks for the suggestion. We updated Table 1 accordingly.

(28) 399-401 – grammar problem, meaning is lost

**Response#28:**

Thanks for pointing that out. The sentence has been modified to "In order to explore the effect of intra-annual rainfall variability on N dynamics, the linear regression analyses between the parameters of the stochastic rainfall generator and N loads, fluxes, as well as  $C_Q$  were conducted, in which the parameters of NY serve as a reference" (Lines 467-470).

(29) 458 – Why is soil denitrification lumped with GW denitrification? Are they expected to be similar?

**Response#29:**

Thanks for point that out. Soil denitrification and GW denitrification are identical in mechanism, and we expected to see how much nitrogen in total is consumed by denitrification under different rainfall pattern. Therefore, they are added together in the study.

(30) 503 – These figures are confusing because the response variables (SON, SIN, LEA, Cq) are not on the z-axes.

**Response#30:**

Thanks for the suggestion! Figure 9 and Figure 10 were presented in the same way. In the stochastic rainfall generator, the distributions of the average storm duration and the average inter-storm period over the course of a year are depicted by Sine functions (Equation S3 & S4, Supporting Information), whose characteristics are determined by the amplitudes of the seasonal variations in the average storm duration and the average inter-storm period ( $\alpha_{\gamma}$  and  $\alpha_{\delta}$ ) reflecting climate change, when other parameters remain constant. We will use Figure 9 as an example. As for a certain  $\alpha_{\gamma}$ , there is a specific distribution of the average storm duration over the course of a year, which is presented in the x-z plane. The color of the panels represents the value of  $\alpha_{\gamma}$ . With  $\alpha_{\gamma}$  decreasing, the distribution of the average storm duration transitions and the year shifts from wet to dry. When the year becomes drier, the annual SON load increases, while the SIN load, leaching flux, and in-stream nitrate concentration decrease on the y-axis. Therefore, the transformation and transport of N are subject to retardation in a dry year when storms with shorter duration occur mid-year.

(31) 564: 5.1 section title: consider being more specific about what increased rainfall does to the N dynamics

**Response#31:**

Thanks for the suggestion. We changed section tile of 5.1 into "*N transformation and transport upgrade in wet years*" (Lines 745-746).

(32) 608: It seems notable that the high flows during the 2018 to 2019 drought are as high as the high flows from 2014-2018, and the main difference during 2018-2019 seems to be in the low flow periods.

**Response#32:**

Thanks for pointing that out. Yes, as you observed, droughts mainly occurred in the middle of the year, resulting in extreme low flow (Figure 14). During the low-flow periods, nitrogen absorbed by vegetation in the growth stage is significantly reduced and nitrogen (SON & SIN) starts to accumulate in the soil. The accumulated nitrogen was subsequently flushed out during the rewetting period [Winter et al., 2023].

**Reference:**

Winter, C., Nguyen, T., Musolff, A., Lutz, S., Rode, M., Kumar, R., Fleckenstein, J.: Droughts can reduce the nitrogen retention capacity of catchments, Hydrol. Earth Syst. Sci., 27(1):303-318, https://doi.org/10.5194/hess-27-303-2023, 2023.

(33) 670: 5.6. Consider discussing: Data limitations, uncertainty of parameters, model process representations

**Response#33:**

Thanks for the suggestions.

We added the discussion to address the model representation and parameter uncertainty and limitation in data as "The transport model preserves the main pathway for N- $NO_3$  leachate by simplifying complexities of different N pools and transformations via mineralization, leaching, and denitrification within the soil zone. However, the external N input entering directly into inorganic N, the transformation from protected organic N to active organic N, and the loss of organic N via dissolution have not been included in the model, which may lead to miscalculation of

nitrogen load. As a complex biogeochemical process, denitrification is governed by various factors such as temperature, soil moisture content, and SIN load, rather than a first-order decay process. Nonetheless, the current transport model does not account for spatial heterogeneity in denitrification. These simplifications may introduce uncertainties in the simulated results. In addition, due to the lack of groundwater nitrate concentration data, the nitrate transport model was calibrated using only in-stream nitrate concentration data [Wang et al., 2023]. More nitrate concentration data in groundwater and river will contribute to establishing a more realistic nitrate transport model" (Lines 905-919).

(34) Table S1 – van Meter reference is missing date; bibliography is not included in this document

**Response#34:**

Thanks for pointing that out. We have corrected the table S1:

**Table S1**. The parameters for the N source zone. The parameters were adjustable and calibrated (referring to Yang et al. [2022]).

| Parameter   | Process                  | Adjustable range                  | Reference               | Value of best fit        |
|-------------|--------------------------|-----------------------------------|-------------------------|--------------------------|
| $k_a$       | Mineralization           | 0~0.7                             | Van Meter et al. [2017] | 0.0111 day -1 |
| $k_p$       | Mineralization           | 0~0.7                             | Van Meter et al. [2017] | 0.0008 day -1 |
| p 1  | Plant uptake             | 60~160                            | Van Meter et al. [2017] | 160 kg·ha -1  |
| p 2  | Plant uptake             | 1~10                              |                         | 10 kg·ha -1   |
| p 3  | Plant uptake             | 1~60                              |                         | 34 day                   |
| $\lambda_s$ | Denitrification in soil  | 0~0.7                             | Nguyen et al. [2021]    | 0.0007 day -1 |
| $k_l$       | Leaching                 | 1~1000                            |                         | 18.8888 day-1            |
| $q_{ref}$   | Leaching                 | 1e -4 ~1               |                         | 0.01 m·day -1 |
| λ           | Denitrification in water | 1e 5 ~1e -1 | Nguyen et al. [2021]    | 0.0088 day-1             |

**RC2:**

The authors present an original piece of research focusing on the catchment response of N fluxes to rainfall interannual and intra-annual variability using synthetic experiments based on the Hydrogeosphere model. I found the study clean and rigorously described, calibration method is sensible. Figures provide useful and clear illustrations of the results. The manuscript is well written. I think the discussion could be expanded and therefore I recommand a minor revision of the manuscript.

**Especially I would have **2 comments on the discussion**:**

1) several parts of the discussion present some reactions as simulation results when they are a direct consequence of the modelling equations. Ex.: Lines 566 to 568, the impact of temperature and wetness on mineralization is constructed mathematically in equations 1 and 2, isn't it? Same comment regarding plant uptake and denitrification. According to my opinion, the interest of the model is rather to calculate which of these mechanisms is going to dominate the response, and also it helps to consider different time scales of response, which is particularly relevant for droughts (cf. lines 97-99)

**Response #1:**

Thank you for the comment. We truly agree that the reaction is a direct consequence of the mechanisms that are built in the equations already. That mean, some trends / patterns can be simply known even without actually performing the simulations. This study is aimed to investigate how the rainfall variability influences the N transport and which process is the key control. E.g., we found that the plant-uptake is the main factor caused the high in-stream concentration in extreme dry years.

In response to your comment, we added the sentence in the result section "Based on these results, it can be preliminarily concluded that annual precipitation and vegetation state critically influence N transformation and transport" (Lines 590-591).

In addition, we have revised the discussion to clarify the dominant mechanisms governing in-stream nitrate concentration responses as "The comparison of  $C_Q$ , N loads, and fluxes across four scenarios (WY, NY, DY, and EDY) reveals the effect of inter-annual rainfall variability on N transformation and transport, as well as water quality. Mineralization, the crucial process of the transformation from SON to SIN, exhibits strong soil moisture dependence. Consequently, the highest average mineralization rates in the WY scenario promoted N transformation through plant uptake, denitrification, and leaching. In contrast, low mineralization in the DY scenario led to SON accumulation, thereby restraining overall N dynamics" (Lines 747-754).

2) I have a general comment on the modelling choice that is N mechanisms are much more simplified than the water processes. While I am very aware of the computationnal challenges associated with such virtual experiments, I find intellectually disturbing to have a fully mechanistic approach to represent water combined with a representation of nitrogen very simplified in comparison. What do you think?

**Response #2.1:**

Thank you for this comment! We totally agree with your point that the N processes represented in the model is very simple compared with the groundwater flow. The complexities N fluxes in source zone were simplified by defining a framework describing the main pathway for N-NO3 leachate with temporally constant external N input. On the one hand, this simplification neglected other processes such as the time-variant external N input, direct input of external N to inorganic N pool, the transformation from protected organic N to active organic N, and the loss of organic N via dissolution. On the other hand, this simplification allows us to focus on the main source of N-NO3-, rather than keeping track of the full nitrogen fluxes in the source zone of the catchment while maintaining the overall nitrogen balance using surplus as a constraint. In this sense, we think it is an effective tool to answer certain questions, for example in our study, which process controls the high nitrate

concentrations in river while dry climate.

To response to this point, we discussed the limitation of the simplified N framework additionally in section 5.7 as follow: "The transport model preserves the main pathway for N- $NO_3$ " leachate by simplifying complexities of different N pools and transformations via mineralization, leaching, and denitrification within the soil zone. However, the external N input entering directly into inorganic N, the transformation from protected organic N to active organic N, and the loss of organic N via dissolution have not been included in the model, which may lead to miscalculation of nitrogen load" [Lines 905-910].

**Figure 2.** The framework simulating the transformation and transport of nitrogen in soil N source zone and groundwater zone, modified based on [Yang et al., 2021].

There is no representation of plant-soil processes, and considering the results of the EDY scenario in terms of N dynamics, the response of plants growth or death to water stress seems to be a key mechanism.

**Response #2.2:**

Thanks for pointing that out! Yes, the plant-soil process is not represented in the model. This is, in the extreme dry scenario, the plants death was not actually

simulated under low soil content, but manually being assigned a lower plant uptake potential. The plant growth stages were assumed to be constant (for example, the same N uptake potential in different years). This is also due to lack of representation for the plant-soil processes in the simplified N framework. N uptake process is only mathematically described using the empirical formula (Equation 6). However, such simplification was proved to be effective in term of reproducing the reasonable N loads, fluxes and in-stream nitrate concentrations. It can still be used to identified the key effect of plant-die-off (i.e. reduced N uptake potential) on stream water quality during extreme dry climate, comparing with other process such as mineralization and leaching.

To response to this point, we discussed the limitation of neglecting the plant-soil processes in section 5.7 as follow: "The plant-soil process was not represented in the model. In the extreme dry scenario, plant death was not actually simulated under high temperature and low soil moisture content, but was manually assigned a lower plant uptake potential. The plant growth stages were assumed to be constant. This is because the N uptake process was only mathematically described using the empirical formula (Equation 6) in the simplified N framework, rather than using a full plant-soil process. However, such simplification was proved to be effective in terms of reproducing reasonable N loads, fluxes and in-stream nitrate concentrations. Therefore, we think it is acceptable to identify the key effect of plant die-off (i.e. reduced N uptake potential) on stream water quality during extreme dry scenarios" (Lines 924-934).

Also, about the fact that all external nitrogen inputs are introduced in the SON pool (lines 250, 252), I was wondering to what extend it refers to a reality? Are fertilizers mainly applied as urea?

Response #2.3:

Thanks for pointing that out!

The external N input represents atmospheric deposition, biological fixation, animal manure from the pasture area, and N fertilizer from the farmland. It does include

organic and inorganic inputs. In our framework based on ELEMeNT model [Van Meter et al., 2017], it is assumed that all external nitrogen inputs are introduced in the organic pool. This assumption is made based on the fact that most of the nitrate (N-NO3-) fluxes from source zone has undergone biogeochemical transformation in organic N pool [Haag and Kaupenjohann, 2001]. The described framework simplifies complexities of different N pools and transformations via mineralization, dissolution, and denitrification within the soil zone, while preserving the main pathway for N-NO3 leachate. In this sense, assumption that the external N input contributes only to the SON is acceptable in the study.

To response to this point, we discussed the limitation of the simplified N framework additionally in section 5.7 as "The transport model preserves the main pathway for  $N-NO_3$  leachate by simplifying complexities of different N pools and transformations via mineralization, leaching, and denitrification within the soil zone. However, the external N input entering directly into inorganic N, the transformation from protected organic N to active organic N, and the loss of organic N via dissolution have not been included in the model, which may lead to miscalculation of nitrogen load. As a complex biogeochemical process, denitrification is governed by various factors such as temperature, soil moisture content, and SIN load, rather than a first-order decay process. Nonetheless, the current transport model does not account for spatial heterogeneity in denitrification. These simplifications may introduce uncertainties in the simulated results. In addition, due to the lack of groundwater nitrate concentration data, the nitrate transport model was calibrated using only in-stream nitrate concentration data [Wang et al., 2023]. More nitrate concentration data in groundwater and river will contribute to establishing a more realistic nitrate transport model" (Lines 905-919).

Also, I did not understand why is this framework acceptable and according to which criteria or arguments (line 258-260)?

Response #2.4:

Thanks for the comment!

Although the mechanisms of N dynamics in this framework are quite simplified, its acceptability is supported by the following points:

- It is effective in term of reproducing the reasonable in-stream nitrate concentrations with higher NSE in the entire catchment. We added the calibration results in the beginning of the section 4 as follows: "The calirbated N transport model showed good performance in fitting the in-stream nitrate concentration (Figure 4a), with a Nash-Sutcliffe efficiency (NSE) of 0.79. The modeled N surplus of 51.87 kg ha-1 yr-1 is comparable to the measured value of 48.8 kg ha-1 yr-1. The calibrated best-fit values for the transport parameters are listed in Table S1 (see Supporting Information)" (Lines 487-491).
- The described framework simplifies complexities of different N pools and transformations via mineralization, dissolution, and denitrification within the soil zone, while preserving the main pathway for N-NO3- leachate. In this sense, the framework can still be used to explore the influence of rainfall on water quality via the key processes (mineralization, plant-uptake and leaching).

To response this point, we briefly compared the simulated N loads and fluxes with observations or data from other literatures in the beginning of the section 4 as follows: "Figure 3 illustrates the 14-year N mass balance simulated by the calibrated N transport model in the entire catchment. In the soil source zone, the total N consisted of SON (552 kg ha-1, including SON(a) of 90 kg ha-1 and SON(p) of 462 kg ha-1) and SIN (48 kg ha-1). The load of SON accounts for 92% of the total N, which corresponds to the research result that the organic N fraction is greater than 90% [Stevenson., 1995]. As for N transformation, the mineralization rate of 173 kg ha-1 yr-1 is within the range (14-187 kg ha-1 yr-1) reported by Heumann et al. [2011] for study sites located in Germany. 71% of the SIN was absorbed by plants at a rate of 123 kg ha-1 yr-1, which is very close to the value (around 120 kg ha-1 yr-1) suggested in Nguyen et al. [2021] for the same area. The denitrification flux of 38 kg ha-1 yr-1 (4 kg ha-1 yr-1 in the soil source zone, 34 kg ha-1 yr-1 in the groundwater) is within the range (8-51 kg ha-1 yr-1) investigated for 336 agricultural areas around the world by Hofstra and

Bouwman [2005]. The SIN entered the groundwater zone at a rate of 44 kg ha-1 yr-1, which is within the range (15-60 kg N ha-1 yr-1) reported in Nguyen et al. [2021]. Approximately 23% of the leachate of SIN was exported to the stream, where the rest of the leachate (77%) was removed by denitrification. The export and denitrification fluxes are comparable to the values 39% and 61% reported in Nguyen et al. [2021]. Based on measured groundwater chemistry data from nearby catchments and multiple estimates of denitrification potential that have been proved and reported [Hannappel et al., 2018], we conclude that the simulated transformation and transport of N are acceptable.

Figure 3. The simulated 14-year N mass balance in the entire catchment." (Lines 492-514).

**I have also a few minor comments below:**

3) line 41: I suspect that fertilization strategies will evolve with the changes in timing of plant growth stages with warming

**Response #3:**

Thanks! This is a very good point. We agree that the fertilization strategies may evolve with warming. Due to the lack of spatiotemporal variation information of the

external N input, its value was fixed at 180 kg ha-1 year-1 according to Nguyen et al. [2021], where the N balance was simulated in the upper Selke catchment covering the Schäfertal catchment. In addition, in this study, the plant growth stages were assumed to be the same for different years (for example, the same sowing date in either wet year or dry year). This is due to lack of representation for the plant-soil processes in the model. Such assumption was proved to be effective in term of reproducing the reasonable N loads, fluxes and in-stream nitrate concentrations [Yang et al., 2021]. However, these assumptions may lead to uncertainties in the simulated N fluxes which is discussed in section 5.7 as following: "The plant-soil process was not represented in the model. In the extreme dry scenario, plant death was not actually simulated under high temperature and low soil moisture content, but was manually assigned a lower plant uptake potential. The plant growth stages were assumed to be constant. This is because the N uptake process was only mathematically described using the empirical formula (Equation 6) in the simplified N framework, rather than using a full plant-soil process. However, such simplification was proved to be effective in terms of reproducing reasonable N loads, fluxes and in-stream nitrate concentrations. Therefore, we think it is acceptable to identify the key effect of plant die-off (i.e. reduced N uptake potential) on stream water quality during extreme dry scenarios" (Lines 924-934).

4) introduction should maybe be restricted to context elements that are directly linked to the study, lines 45 to 61: it could go more directly to the point of rainfall variability skipping the details on extremes. Lines 79 to 82: I missed the link with the present study here.

**Response #4:**

Thank you for your suggestions. We have revised the first paragraph of the Introduction as "The hydrological processes are susceptible to meteorological conditions on various spatial and temporal scales [Ionita et al., 2017; Laaha et al., 2017; Zhang et al., 2021]. In the past decades, extreme climate events intensified by human-induced climate change have frequently occurred globally [Pall et al., 2011;

Min et al., 2011; Williams et al., 2015; Hari et al., 2020], most of which caused water scarcity and poor water quality at regional scales [Zwolsman and van Bokhoven, 2007; Delpla et al., 2009; Whitehead et al., 2009; Stahl et al., 2016; Ballard et al., 2019; Bauwe et al., 2020; Geris et al., 2022]. Heavy rainstorms and severe droughts being the predominant extreme climate events around the globe share the common characteristic of rainfall variability [Trenberth et al., 2011; Pendergrass et al., 2017; Hanel et al., 2018]. In the context of global warming scenarios, anthropogenic amplification of rainfall variability has been identified [Zhang et al., 2024]. Thus, the effect of rainfall variability on water resources has attracted much attention around the world? (Lines 59-80).

We also deleted the following sentences describing historical extreme events, which are not directly relevant to this study "Extreme rainfall events in 2002, 2013, and 2021 caused significant threats to human safety and substantial damage to the environment, economy, and infrastructure [Ulbrich et al., 2003; Thieken et al., 2016; Voit & Heistermann, 2024]. Three notable summer droughts occurred in 2003, 2015, and 2018–2019 (consecutive), driven by precipitation deficits combined with temperature anomalies during the growing season [Fink et al., 2004; Schär et al., 2004; Ciais et al., 2005; Orth et al., 2016; Hanel et al., 2018; Hari et al., 2020; Camenisch et al., 2020]" (Lines 104-111).

5) lines 145-146: at which frequency are concentration measured?

**Response #5:**

Thanks! We clarified this by adding "..at 14-days to monthly intervals [Dupas et al., 2017], covering the period 2001–2010" (Lines 186-187).

6) line 339-340 what is a "good performance under Climate change" for the stochastic generator?

**Response #6:**

Thank you for raising this point.

Wilusz et al. [2017] coupled the rainfall generator [Robinson & Sivapalan, 1997], a rainfall-runoff model [Kirchner, 2009], and the rank StorAge Selection (rSAS) transit time model [Harman, 2015] to decompose the relationship between rainfall variability and the time-varying fraction of young water (

**Figure 1.** (a) The catchment 'Schäfertal' indicated by the red line (background image from © Google Maps), with a cross-sectional view for the flow and saturation [Yang et al., 2018]. (b) The distribution of soil type in the catchment. (c) The measured daily precipitation (P), discharge (Q), and the simulated actual evapotranspiration (ET) [Yang et al., 2018].

9) it would be useful to see the effect on water fluxes as well: especially recharge flux for groundwater and Actual ET in Figure 10 or Figure S2 (lines 653 to 656)

**Response #9:**

Thank you for your suggestion. We added the actual ET and the rechange fluxes (Rainfall-actual ET) in Figure S2 (see Supporting Information).

---

## Author Response (AR2)

Dear reviewers and editor,

Thank you so much for your constructive comments. Please check our responses to all your comments point by point (followed and marked in blue). Unless specified, all the line numbers in our responses refer to the line numbers of the TRACK CHANGES

version of the manuscript.

Besides, the responses to the two reviewers, we also made the following updates to

the manuscript:

1. We acknowledged the work of editorial board and reviewers in the

acknowledgements (Lines 834-836).

2. We updated the author's affiliation (Lines 11-13) and "Financial support"

(Lines 837-846).

3. We updated Figure 13 (Lines 654-656) to maintain consistency with the format

used in the other figures. We have made minor language and grammar edits

throughout to improve readability; these changes do not affect the scientific

content.

**RC1:**

Line-by-line comments:

18: "aims to investigate"

Response#1:

Thanks for pointing that out. We revised it (Line 19).

23-24: "Scenarios are high, normal, low, and extremely low annual"

Response#2:

Thanks for the suggestion. We modified it into "...four scenarios (wet, normal, dry and extremely dry conditions)..." (Lines 24-26).

38: "a drought"

Response#3:

Thanks for pointing that out. We revised it (Line 40 & 807).

41: delete "merely"

**Response#4:**

Thanks for the suggestion. We deleted it (Line 43 & 810).

Line 57: "Heavy rainstorms and severe droughts being the predominant..." -> "Heavy rainstorms and severe droughts, the predominant..."

**Response#5:**

Thanks for pointing that out. We rephrased this sentence as "As predominant extreme climate events worldwide, heavy rainstorms and severe droughts share a common characteristic of rainfall variability" (Lines 59-61).

Line 87: "detected higher soil N surplus ... and decreased the terrestrial N export" -> "detected higher soil N surplus ... and a decrease in terrestrial N export".

**Response#6:**

Thanks for pointing that out. We modified it (Line 92).

Line 94: "They seem opposite conclusions" -> "These appear to be opposite conclusions".

**Response#7:**

Thanks for the suggestion. We modified this sentence accordingly (Lines 101-102).

139: "The types of land use in the catchment do not generally convert until the economic and ecological goals vary between years" -> "The land use types generally remain stable unless economic and ecological goals change"

**Response#8:**

Thanks for the suggestion. We corrected it accordingly (Lines 149-152).

146: "is considered as the unique exit" -> "is considered the sole exit"

**Response#9:**

Thanks for the suggestion. We revised it accordingly (Line 155-156).

244: "The transformation and transport of nitrogen in the underground area are tracked…" -> "The transformation and transport of nitrogen in the subsurface are tracked…"

**Response#10:**

Thanks for the suggestion. We modified it accordingly (Line 257).

374: Consider providing values here to put the qualitative scenarios in context. For example, what was the total rainfall in each scenario and how does that value compare to rainfall amounts in historical records?

**Response#11:**

Thanks for this suggestion. We added key information about historical meteorological data and rephased this sentence as "To represent contrasting hydroclimatic conditions, three years were selected from the 1997–2022 record (mean 607.9 mm; range 408.2-916.3 mm): the wet year (2007, P = 916.3 mm), the normal year (2008, P = 588.7 mm), and the dry year (2018, P = 444.1 mm);" (Lines 370-374)

437: Figure 3 – y axis labels are ambiguously positioned. Which bars go with which label?

**Response#12:**

Thanks for pointing that out. We modified the figure, as shown in Figure 3 (Lines 454-455).

*Figure 3.* The simulated 14-year N mass balance in the entire catchment.

448: "transforms" -> "shifts"

**Response#13:**

Thanks for the suggestion. We modified it accordingly (Line 469).

457: "the transformation of SON is not subject to vegetation state" -> "SON transformation is independent of vegetation state".

**Response#14:**

Thanks for the suggestion. We modified it accordingly (Line 479).

473: Readers may be confused why denitrification would be greater during an extremely dry year with low soil moisture.

**Response#15:**

Thanks for pointing that out. Accumulated SIN caused by reduced plant uptake makes denitrification be greater during an extremely dry year with low soil moisture. Thus, the sentence was modified into "Because of the accumulated SIN load (Figure 6c) that resulted from the lowest level of plant uptake, denitrification and leaching fluxes in

*EDY with low annual precipitation are still even larger than those of WY.*" (Lines 492-496).

555 - "In warm periods, enhanced average rainfall intensity increased actual evapotranspiration (ET) (Figure S2b1 and S2c1)," The description seems to imply a positive correlation, but figure S2b1 and S2c1 both show steep negative slopes.

**Response#16:**

Thanks for this question. Negative values indicate that water leaves the subsurface. Figure S2 was modified to display actual evapotranspiration as positive values for clarity.

**Figure S2.** The responses of actual evapotranspiration (ET), recharge for groundwater, discharge (Q), and N export to the average rainfall intensity of the four seasons ( $E_1$ - $E_4$ ). The determination coefficients ( $R^2$ ) between  $E_1$ - $E_4$  and ET, recharge, Q, and export are listed. The sign and magnitude of the slopes (k) in these linear relationships denote the direction and the intensity of response of N dynamics to the variations in average rainfall intensity. Asterisks

indicate the significance of the regression slopes (p < 0.05); ns denotes non-significant relationships ( $p \ge 0.05$ ).

557: "both recharge and discharge increased during the second season (Figure S2b2 and S2b3)." The description seems to imply a positive correlation. On plots S2b2 and S2b3, R-squared values are 0.01 and 0.00, and there are no discernable trends in the data points.

**Response#17:**

Thanks for pointing that out. After remarking the significance of linear regressions (in the next reply), we replaced the inaccurate description as "In warm periods, enhanced average rainfall intensity increased actual evapotranspiration (ET) (Figure S2b1 and  $S2c_1$ ). The linear relationships between seasonal average rainfall intensity and each of recharge, Q, N export, and  $C_Q$  are weak, with low  $R^2$  and small slope (k) values. Although several regressions reach statistical significance (p < 0.05), their effects are minor (Figure 11b7, S2a2, and S2d4)." (Lines 583-591). In addition, we modified the related description about denitrification into "Warmer temperatures and enhanced moisture during the growing season promoted vigorous nutrient absorption by vegetation (Figure 11b5 and 11c5). In addition, soil denitrification increased markedly during the third season as average rainfall intensity increased (Figure 11c6), which was due to favorable microbial conditions. In the fourth season, soil denitrification decreased slightly with increasing average rainfall intensity (Figure 11d6), which was attributed to low temperatures and decreased SIN loads (Figure 11d3)." (Lines 576-583). The related discussion was also revised "However, denitrification is dependent on microbial conditions as well. Leaching is additionally influenced by soil saturation and groundwater velocity (Equation 3 and 7). As a result, no significant linear relationship exists between mean rainfall intensity and leaching. It is noteworthy that high-intensity precipitation events with short durations and substantial surface runoff rarely reach the water table, thereby exerting minimal effects on recharge, discharge, and N export (a component of leaching flux) (Figure S2, Supporting Information). Therefore, Co shows weak and non-significant responses to extreme precipitation." (Lines 732-742).

563: Consider including information about which of the trends are statistically significant.

**Response#18:**

Thanks for this suggestion. We remarked the significance of linear regressions in Figure 11, 12, and S2. The method and results of linear regressions were added into the related

paragraphs (Lines 408-412, 527-528, 533-534, 549-551, 568-569, 615-617).

Figure 11. The responses of N loads, mineralization, plant uptake, and denitrification (in soil) fluxes, as well as in-stream nitrate concentration  $(C_Q)$  to the average rainfall intensity of the four seasons  $(E_1-E_4)$ . The sign and magnitude of the slopes (k) in these linear relationships denote the direction and intensity of the response of N dynamics to the variations in average rainfall intensity, respectively. Asterisks indicate the significance of the regression slopes (p

Figure 12. The responses of N loads, fluxes, discharge (Q) and in-stream nitrate concentration ( $C_Q$ ) to the probability of drizzle events ( $P_{drizzle}$ ). The sign and magnitude of the slopes (k) in these linear relationships denote the direction and intensity of the response of N dynamics to the probability of drizzle events ( $P_{drizzle}$ ), respectively. Asterisks indicate the significance of the regression slopes (p

**Figure S2.** The responses of actual evapotranspiration (ET), recharge for groundwater, discharge (Q), and N export to the average rainfall intensity of the four seasons ( $E_1$ - $E_4$ ). The determination coefficients ( $R^2$ ) between  $E_1$ - $E_4$  and ET, recharge, Q, and export are listed. The sign and magnitude of the slopes (k) in these linear relationships denote the direction and the intensity of response of N dynamics to the variations in average rainfall intensity. Asterisks

indicate the significance of the regression slopes (p < 0.05); ns denotes non-significant relationships ( $p \ge 0.05$ ).

642: "induce prominent changes"

Response#19:

Thanks for the suggestion. We deleted "the" accordingly (Line 681).

677: ""rarely propagate to the groundwater zone, minimally affecting discharge and nitrate export" -> "rarely reach the water table and thus minimally affect discharge and

nitrate export".

Response#20:

Thanks for the suggestion. The sentence was modified into "It is noteworthy that high-intensity precipitation events with short durations and substantial surface runoff rarely propagate toreach the groundwater zonewater table, thereby exerting, minimally aeffects oning recharge, discharge, and nitrate N export (a component of leaching flux) (Figure S2, Supporting Information)." (Lines 716-720).

688: "More discharge yielded in extreme dry-wet patterns than in continuously humid conditions" -> "Extreme dry-wet patterns yielded more discharge than continuously humid conditions".

Response#21:

Thanks for the suggestion. This sentence and the next have been removed based on the context of the surrounding text (Lines 730-733).

705: "miscalculations"

Response#22:

Thanks for pointing that out. We modified it (Line 750).

709: delete "nonetheless"

**Response#23:**

Thanks for the suggestion. We deleted it (Line 753).

729: "Abundant microorganisms and animals engage in extensive activities, and massive organic and inorganic matter undergoes biochemical reactions ··· " -> "Microorganisms and soil fauna are highly active, and soil organic and inorganic matter undergo continual biochemical reactions ··· ".

**Response#24:**

Thanks for the suggestion. We modified it accordingly (Line 772-775).

737: "salt clusters" -> "salt accumulation"

**Response#25:**

Thanks for the suggestion. We modified it accordingly (Line 782).

751: delete "nonetheless"

**Response#26:**

Thanks for the suggestion. We deleted it (Line 29 & 797).

**RC2#:**

**line 25 remove interannual**

**Response#1:**

Thanks for this suggestion. We prefer to retain "inter-annual" (Line 27), which corresponds to "intra-annual" (Line 34). They denote the timescales of experimental rainfall variability.

line 29-30 of course, vegetation plays a vital role on N dynamics! I would rephrase as "vegetation response to extreme droughts will be the main controller of N fluxes

response to these extreme climatic events"

Response#2:

Thanks for pointing that out. We modified the sentence into "vegetation plays a vital role in the response of N dynamics to extreme droughts" (Line 31-32 & 799-800).

line 57-59 change for "heavy rainstorms and severe droughts are the predominant extreme climate events around the globe and are associated with change in rainfall variability"

Response#3:

Thanks for pointing that out. We rephrased this sentence as "As predominant extreme climate events worldwide, heavy rainstorms and severe droughts share a common characteristic of rainfall variability" (Lines 59-61).

line 67-68: "On the opposite, severe droughts driven by precipitation deficits occur during several months and potentially years and it takes 1-2 years for hydrological components to recover"

Response#4:

Thanks for your suggestion. The sentence was modified into "Different from heavy rainstorms, severe droughts driven by precipitation deficits occur during several months and potentially years [Otkin et al., 2018], from which it takes 1-2 years for hydrological components to recover [Hanel et al., 2018]." (Lines 70-73)

line 88: remove "the" terrestrial ....

Response#5:

Thanks for pointing that out. We deleted it (Line 92).

line 101-106: the framing of knowledge gap could be improved here: I think that the scope of your study is to move beyond the case study shedding light on the effects of rainfall variaiblity on N dynamics in a more systematic way and investigating the underlying mechanisms using a synthetic experimental study

Response#6:

Thanks for the suggestion. The framing of knowledge gap was rephrased as "...Zhou et al. [2022] detected higher soil N surplus (total N input with the crop/plant uptake subtracted) and a decrease in terrestrial N export in agricultural areas located in

Central Germany during the drought years (2015-2018). The same phenomenon reported in the Nitrate Report 2020 of the Netherlands (RIVM, 2021) indicates that more N was retained in the soil during the drought period compared to the pre-drought period. Notably, the 2018–2019 (consecutive) drought triggered unprecedented tree mortality across multiple species in Central European forests, accompanied by unexpectedly persistent drought legacy effects [Schuldt et al., 2020], from which Winter et al. [2023] drew the conclusion that severe multi-year droughts can reduce the nitrogen (N) retention capacity of catchments. These appear to be opposite conclusions, which can be attributed to different investigation timescales. The former study compared N export between drought years and the pre-drought period. The latter considered the subsequent rewetting period, when most nitrogen accumulated during the drought left the catchment. Leitner et al. [2020] also found that in the year after a summer drought, NO3- leaching via soil water seepage was significantly elevated compared to the long-term mean in a temperate mixed forest on karst, which was investigated in wetland-influenced catchments as well [Watmough et al., 2004]. These studies demonstrate that rainfall variability profoundly affects N dynamics at both inter-annual and intra-annual timescales. Therefore, it is imperative to shed light on the impact of rainfall variability on water quality in terms of N dynamics.

To fill the gap, the present study explored the impact of rainfall variability on N dynamics and its potential influence on water quality across inter-annual and intra-annual timescales." (Lines 88-116).

line 121: specific parameters of rainfall distribution were used. Response#7:

Thanks for your suggestion. We modified the sentence into "...rainfall time series generated by separately altering specific rainfall-generator parameters substitute for the rainfall data in the simulation period to drive the flow and nitrogen transport models." (Lines 130-133).

lines 716-725 Related to my question on plant processes, another point is that in a very dry year, it is also very likely that farmers will modify their fertilization. Nevertheless the input is contant here in the model isn't it?

**Response#8:**

Thanks for the insightful comment. We agree that farmers may adapt fertilization in very dry years. Due to limited information on farm behavior and model limitation,

considering the adaptive farm management is outside the scope of this work. The present study is designed to explore the effects of rainfall variability on nitrogen (N) dynamics. To achieve the attribution, we intentionally held temporally constant fertilization rate across scenarios and represented extreme drought by very low annual precipitation and reduced plant uptake potential due to vegetation dieback. This design avoids confounding management decisions with hydrometeorological drivers and allows us to focus on mechanism identification (plant uptake, denitrification, and leaching) under rainfall variability. We have clarified this assumption in the Methods "To isolate the causal effects of rainfall variability on N dynamics, the time-invariant fertilization rate was used across all scenarios" (Lines 396-398). We believe this scoping choice is appropriate for the study's objective.